🔓 | **Open Peer Review** | Computational Biology | Research Article

# TolRad, a model for predicting radiation tolerance using Pfam annotations, identifies novel radiosensitive bacterial species from reference genomes and MAGs

Philip Sweet,[1] Matthew Burroughs,[1] Sungyeon Jang,[1] Lydia Contreras[1]

**ABSTRACT** The trait of ionizing radiation (IR) tolerance is variable between bacterium, with species succumbing to acute doses as low as 60 Gy and extremophiles able to survive doses exceeding 10,000 Gy. While survival screens have identified multiple highly radioresistant bacteria, such systemic searches have not been conducted for IR-sensitive bacteria. The taxonomy-level diversity of IR sensitivity is poorly understood, as are genetic elements that influence IR sensitivity. Using the protein domain (Pfam) frequencies from 61 bacterial species with experimentally determined $D_{10}$ values (the dose at which only 10% of the population survives), we trained TolRad, a random forest binary classifier, to distinguish between radiosensitive ($D_{10} < 200$ Gy) and radiation-tolerant ($D_{10} > 200$ Gy) bacteria. On untrained species, TolRad had an accuracy of 0.900. We applied TolRad to 152 UniProt-hosted bacterial proteomes associated with the human microbiome, including 37 strains from the ATCC Human Microbiome Collection, and classified 34 species as radiosensitive. Whereas IR-sensitive species ($D_{10} < 200$ Gy) in the training data set had been confined to the phylum *Proteobacterium*, this initial TolRad screen identified radiosensitive bacteria in two additional phyla. We experimentally validated the predicted radiosensitivity of a *Bacteroidota* species from the human microbiome. To demonstrate that TolRad can be applied to metagenome-assembled genomes (MAGs), we tested the accuracy of TolRad on Egg-NOG assembled proteomes (0.965) and partial proteomes. Finally, three collections of MAGs were screened using TolRad, identifying further phyla with radiosensitive species and suggesting that environmental conditions influence the abundance of radiosensitive bacteria.

**IMPORTANCE** Bacterial species have vast genetic diversity, allowing for life in extreme environments and the conduction of complex chemistry. The ability to harness the full potential of bacterial diversity is hampered by the lack of high-throughput experimental or bioinformatic methods for characterizing bacterial traits. Here, we present a computational model that uses *de novo*-generated genome annotations to classify a bacterium as tolerant of ionizing radiation (IR) or as radiosensitive. This model allows for rapid screening of bacterial communities for low-tolerance species that are of interest for both mechanistic studies into bacterial sensitivity to IR and biomarkers of IR exposure.

**KEYWORDS** ionizing radiation, metagenomics, bioinformatics, human microbiome, genome analysis, *Bacteroides*, microbiome, random forest, oxidative stress, marine microbiology

Advances in sequencing technology and genome assembly algorithms have led to an exponential increase in the number and diversity of available bacterial genomes (1). Third-generation sequencing platforms and single-cell sequencing methods can capture an even greater genetic diversity of bacterial communities (1). These new

Address correspondence to Lydia Contreras, lcontrer@che.utexas.edu.

The authors declare no conflict of interest.

See the funding table on p. 16.

methods do not require the ability to isolate and culture individual strains within a community, eliminating a barrier that previously limited the diversity of bacteria that could be studied (2). While there are still challenges to assembling metagenomes (3), current methods have allowed for the description of thousands of new bacterial genomes, such as from ocean water (4), the human microbiome (5), soil (6), and even the international space station (7). This rise in genomic data highlights the need for computational tools for characterizing bacterial traits. Experimentally established traits, such as respiratory preference, gram stain, carbon source utilization, and antibiotic tolerance, have historically been used to describe new bacterial species (8); however, these established methods are not suited for community-level analysis. Experimentally determining the traits of entire communities of bacteria is impractical; yet, to understand how complex communities of bacteria are interacting, knowing which traits are associated with which species is essential (3).

To aid in the interpretation of novel bacterial genomes, predictive algorithms have been developed that can generate genome annotations directly from genomic sequences. Genome annotation tools can identify protein-coding regions (Prodigal) (9), determine protein structure (AlphaFold2) (10), and suggest possible protein functions (Pfam) (11). These tools allow for the description of individual proteins, but the connection between collections of proteins and specific phenotypic traits is still poorly understood. In response to these issues, statistical models that connect genome annotations to phenotypic traits have been developed. Models have been written to predict traits such as the nature of prophages (BACPHLIP) (12), metabolic preference (13), virulence (14), and antibiotic tolerance (15, 16). Often, these models use Pfam domains, annotations assigned to protein sequences using a set of hidden Markov models (11), to inform the classification. The ability to computationally predict bacterial tolerance for stress from such genome annotations is currently limited to antibiotic resistance (15, 16).

There is a renewed interest in understanding native bacterial tolerance for ionizing radiation (IR). IR is a complex stress that threatens the stability of the bacterial genome. A greater understanding of bacterial sensitivity to IR has implications for diverse research topics, including the hardening of bacteria against IR for bioremediation of radioactive sites (17), developing bacterial biomarkers of IR exposure (18), and understanding the risk posed to the human microbiome by space flight (19) and radiation therapy (20). For example, a recent metareview of the response of the human microbiome to radiation noted changes in species diversity and abundance after radiation exposure but highlighted the lack of clarity around the susceptibility of these species to IR (21).

Exposure of bacterial cells to IR causes both direct damage when ionized particles collide with macromolecules (22) and indirect damage generated by reactive oxygen species (ROS) resulting from radiolysis. DNA damage (23) and protein oxidation (24) have both been observed after the exposure of cells to IR. To compare IR tolerance between bacterial species, the acute dose at which only 10% of the exposed cells will produce viable colonies ($D_{10}$) is used. The tolerance of bacteria for acute doses of IR varies greatly between species (25). For instance, the extremophile *Deinococcus radiodurans* (*D. radiodurans*) has a $D_{10}$ of 12,000 Gy whereas the IR-sensitive bacterium *Shewanella oneidensis* (*S. oneidensis*) has a $D_{10}$ of 70 Gy (26). Determining the $D_{10}$ of a bacterium requires access to a powerful irradiation source and the ability to individually culture the bacterium in question. These demands have limited the number of bacterial species for which $D_{10}$ values are available. Historically, the identification of IR-tolerant bacteria has been advanced by the need to understand the dose of IR required for the sterilization of medical compounds (27), food (28), and wastewater (29). Work has also been done to identify extremophiles from off-world analog sites such as the Taklimakan desert (30). These research motives have favored the discovery of radiation-tolerant bacteria over bacteria sensitive to IR. And yet, when trying to understand the disruption of important bacterial communities, such as the gut microbiome (19, 21), that occurs after IR exposure, there is a limited understanding of which species are most sensitive to radiation and what factors make certain bacteria more or less sensitive to IR. In part, this limitation

is because the genetic diversity of bacteria sensitive to IR is poorly understood, with generalizations about radiation tolerance being limited to observations, e.g., gram-negative bacteria are generally less tolerant than gram-positive bacteria (31). Studies of the mechanism behind bacterial sensitivity to IR are hindered by the small number of IR-sensitive bacteria currently known. Experimental characterization has only discovered 14 bacterial species with $D_{10}$ values less than 200 Gy (Table S1), and all these bacteria belong to the *Protobacterium* phylum.

The best-understood predictor of IR tolerance in bacteria is the intracellular ratio of manganese to iron (Mn/Fe) (23). Research comparing the IR-sensitive bacterium *S. oneidensis* and the extremely IR-tolerant bacterium *D. radiodurans* (24, 26) has found that the intracellular ratio of Mn/Fe is correlated with IR tolerance. This ratio has been found to generally be predictive of microorganism IR tolerance, including across five bacterial species (23). The relationship between the intracellular ratio of Mn/Fe and IR tolerance is explained by the opposing effect these ions have on the spread of ROS. Iron furthers the spread of ROS through Fenton chemistry, whereas manganese is an antioxidant, acting as a sponge of ROS. While determining the intracellular ratio of Mn/Fe does not require an IR source, it does require isolated culture growth. Complicating the interpretation of Mn/Fe ratios, the intracellular ratio of Mn/Fe has been shown to be influenced by growth media (24). To date, a correlation between gene or protein frequency and levels of intracellular ratios of Mn/Fe between bacteria has not been determined.

To identify novel IR-sensitive bacterial species, we constructed a model capable of using the frequency of Pfam domains to identify bacterial species with a low survival threshold for IR. In this paper, we describe the construction and validation of the tolerance for radiation (TolRad) model. We also describe the application of TolRad to pre-annotated reference proteomes and metagenome-assembled genomes (MAGs). TolRad is a random forest binary classifier that uses the relative frequency of Pfam annotations within a bacterial proteome to classify a genomic assembly as coming from a bacterium that is tolerant of ($D_{10} > 200$ Gy) or radiosensitive ($D_{10} < 200$ Gy) IR exposure. To build TolRad, a diverse set of 61 bacteria, with associated Pfam annotations and experimentally determined $D_{10}$ values, was split 70/30 into a train set and test set. The final TolRad model utilized four Pfam domains (PF03466, PF07992, PF00300, and PF00849). The accuracy of TolRad on the train set was 0.875. To validate the ability of TolRad to classify species it was not trained on, the model was applied to the test set, on which it was 0.900 accurate.

By applying TolRad to a collection of UniProt-assembled proteomes, 34 species were classified as putative radiosensitive. Of particular interest was the prediction that 19 of the 29 *Bacteroidetes* species, many of which are abundant in the human gut, were classified as radiosensitive. One of the bacteria predicted as radiosensitive was the key human gut commensal *Bacteroides thetaiotaomicron* (*B. thetaiotaomicron*). As no members of the *Bacteroidota* phylum have yet been experimentally characterized as having a low survival threshold to IR, we validated that *B. thetaiotaomicron* indeed had a $D_{10}$ value below 200 Gy. The ability of TolRad to correctly identify radiosensitive bacteria from *de novo* annotated Pfam domains was tested by reannotating the genomes of the train/test set with EggNOG-Mapper (32). TolRad suffered no decrease in accuracy (0.97) on the EggNOG-Mapper (32) annotated genomes. We then applied TolRad to MAGs from three previously published data sets: a set of human microbiome bacteria (HMB) (33), a set of bacteria collected from a glacial stream in the Canadian High Arctic (CHA) (34), and a collection from the deep ocean (35). Broadly speaking, TolRad predicted a greater ratio of putative radiosensitive bacteria from the deep ocean (30.7%) and human gut microbiome (HGM) (7.5%), compared with the CHA (6.9%) and human skin microbiome (HSM) (0%). Screening these MAGs further expanded the diversity of putative radiosensitive species and supports the idea that environmental conditions besides radiation can influence IR tolerance.

In summary, we demonstrate that TolRad, a random forest classifier built using a data set of previously published survival values and genome annotations, can be used to

predict a species' tolerance for IR. A similar workflow, as we describe for the construction of TolRad, could be used to develop a predictive classifier for other environmental stresses. Specifically, using TolRad, we have identified novel species of radiosensitive bacteria from the human microbiome. Broadening the diversity of known radiosensitive bacteria will be vital for future studies into the mechanisms of radiation tolerance and for identifying bacterial biomarkers of radiation exposure.

## RESULTS

### Collection of the train/test set

TolRad was trained and tested on a manually curated data set of 120 experimentally determined bacterial $D_{10}$ values representing 61 species (Tables S1 and S2) and the associated frequency of Pfam domains, referred to as the train/test set. Due to the impact that exposure conditions can have on IR tolerance, only $D_{10}$s determined in liquid media or phosphate-buffered saline (PBS) and exposed at room temperature were included. For all D10s included in the train/test, all exposures were acute and conducted using gamma or X-ray sources. When multiple $D_{10}$ were found, the mean was used (Table S2). The train/test set contains a diversity of bacteria from seven different phyla and includes species that are anaerobic, aerobic, and facultative anaerobic as well as gram-positive and -negative species (Table S1).

### Establishing the IR sensitivity cutoff

Within the train/test set, the $D_{10}$ values varied from 60 to 15,000 Gy, with a heavy rightward skew (Fig. 1A). Bacteria previously described as extremely radiation-tolerant ($D_{10}$ > 1,200 Gy) (Table S1) were found within the rightward tail. We defined the bacteria of the lower quarter of the distribution (Fig. 1A) as radiosensitive, resulting in a cutoff of $D_{10}$ < 200 Gy. The species with $D_{10}$ values between 200 and 1,200 Gy were termed "moderately tolerant." Since we were most interested in identifying radiosensitive species, we combined the "moderately tolerant" and "extremely tolerant" categories into a "tolerant" category ($D_{10}$ > 200 Gy). Using a $t$-test, we demonstrated that, within the train/test set, there was no statistically significant difference ($P$-value <0.05) in optimal growth temperature or the genome GC content between the radiosensitive and tolerant species (Fig. S1). In alignment with previous IR literature (31), none of the gram-positive species were classified as radiosensitive. At the phylogenetic level, all the radiosensitive species were *Proteobacteria*. The represented bacteria from the remaining phyla (*Actinobacteria*, *Aquificota*, *Firmicutes*, *Bacillota, Bacteroidetes,* and *Deinococci*) were classified as tolerant to IR. TolRad was trained specifically to differentiate radiosensitive species ($D_{10}$ < 200 Gy) from tolerant species ($D_{10}$ > 200 Gy) and does not assign relative tolerance, only a binary classification.

### Predictor selection

Based on the success of previous models that have predicted bacterial traits using Pfam domains (12, 36), we started by identifying Pfam domains that correlated with IR tolerance. The train/test set was then randomly split, 70/30, into a train set and a test set (Table 1; Table S1). The train set was used to select predictors of IR tolerance and train the model. The test set was totally hidden from the construction of the model. Of the 7,409 unique Pfam domains present in the 40 proteomes in the train set, 132 were abundant (more than two occurrences per genome). Using the Boruta feature selection algorithm (37), the relative frequency of 7 of these 132 Pfam annotations was found to correlate with IR tolerance classification. Step-wise removal was used to select the most parsimonious model, which resulted in the final selection of four Pfam domains (Table 2). A visualization of the predictor selection pipelines is presented in Fig. 1B. To provide context to the biological relevance of these predictor Pfam domains, we examined the *E. coli* proteins that contain these domains (summarized in Table 2). We also calculated the mean decrease in accuracy (Table 2) for each predictor in the

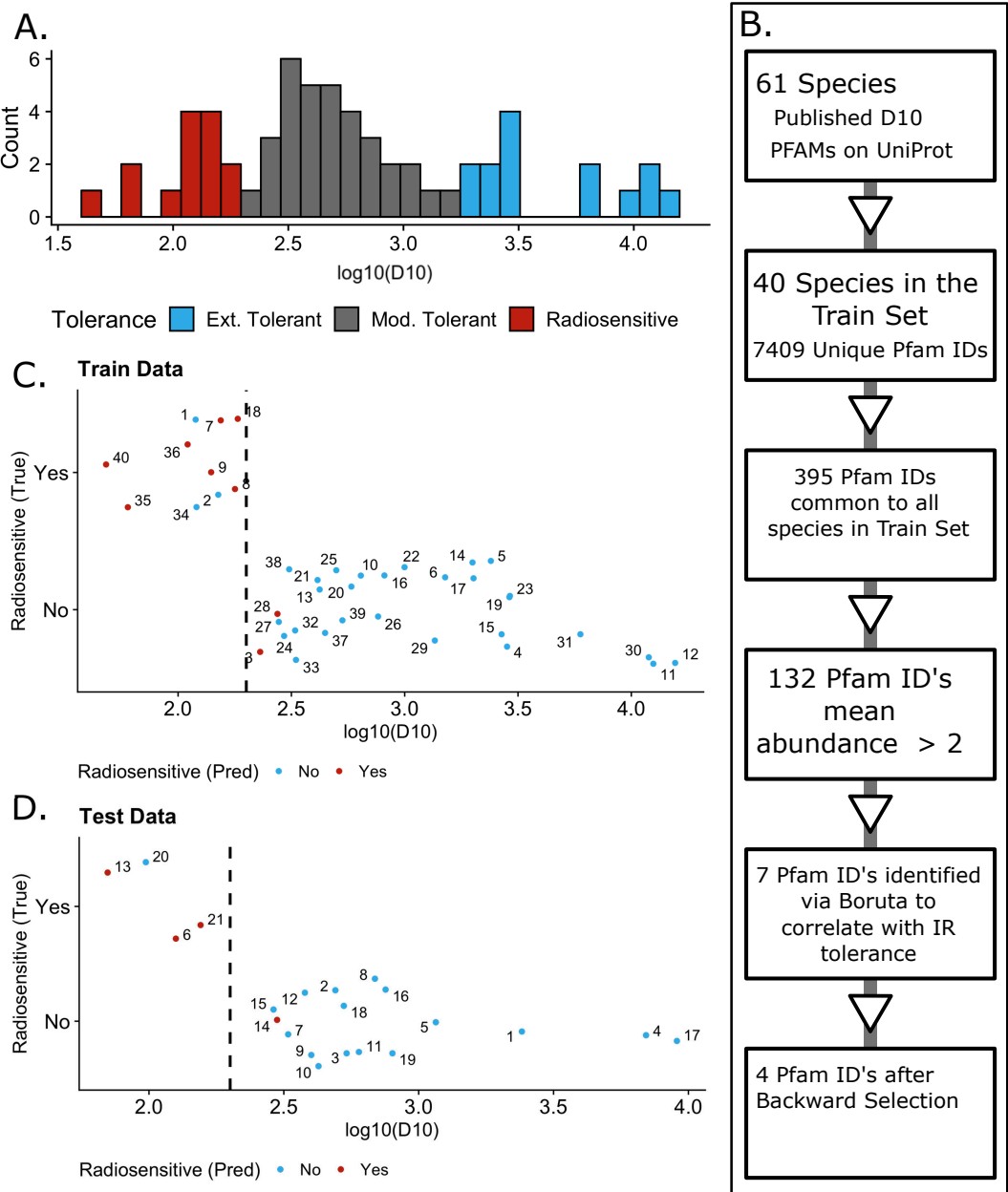

**FIG 1** Construction of TolRad. (A) Histogram distribution of the train/test set. Each count is the mean $D_{10}$ of a unique species. A total of 61 species were used to train the model, representing 121 experimentally determined $D_{10}$ values. (B) Workflow used for training TolRad. (C) Performance of TolRad on the train set. Numbers are a unique species. 1, *Acinetobacter calcoaceticus*; 2, *Aeromonas hydrophila*; 3, *Aeromonas salmonicida*; 4, *Aquifex pyrophilus*; 5, *Bacillus pumilus*; 6, *Bacillus sphaericus*; 7, *Campylobacter coli*; 8, *Campylobacter jejuni*; 9, *Campylobacter lari*; 10, *Coxiella burnetiid*; 11, *Deinococcus geothermalis*; 12, *Deinococcus radiodurans*; 13, *Enterobacter* sp. BIGb0383; 14, *Enterococcus faecium*; 15, *Enterococcus faecalis*; 16, *Escherichia coli*; 17, *Kineococcus radiotolerans*; 18, *Klebsiella variicola*; 19, *Kocuria rhizophila*; 20, *Lactococcus lactis*; 21, *Listeria monocytogenes*; 22, *Methylobacterium radiotoleran*; 23, *Micrococcus luteus*; 24, *Morganella morganii*; 25, *Mycobacterium smegmatis*; 26, *Mycobacterium tuberculosis*; 27, *Proteus vulgaris*; 28, *Pseudomonas putida*; 29, *Rhodococcus erythropolis*; 30, *Rubrobacter radiotolerans*; 31, *Rubrobacter xylanophilus*; 32, *Salmonella Senftenberg*; 33, *Salmonella enterica* subsp. *enterica* serovar Heidelberg; 34, *Salmonella paratyphi*; 35, *Serratia marcescens*; 36, *Shewanella putrefaciens*; 37, *Staphylococcus aureus*; 38, *Staphylococcus epidermidis*; 39, *Stenotrophomonas maltophilia*; 40, *Vibrio parahaemolyticus*. Graph IDs can also be found in Table S3. Color denotes the classification assigned by TolRad. Red denotes a species classified as radiosensitive. Blue denotes a species classified as tolerant. The dashed line at 200 Gy is the cutoff used for defining radiosensitive vs tolerant species. (D) Same as (C), but for the train set classifications. 1, *Acinetobacter radioresistens*; 2, *Bacillus cereus*; 3, *Bifidobacterium breve*; 4, *Deinococcus ficus*; 5, *Lactobacillus acidophilus*; 6, *Neisseria gonorrhoeae*; 7, *Paenibacillus amylolyticus*; 8, *Pediococcus pentosaceus*; 9, *Priestia megaterium*; 10, *Salmonella enterica*; 11, *Salmonella muenster*; 12, *Salmonella typhimurium*; 13, *Shewanella oneidensis*; 14, *Shigella boydii*; 15, *Shigella flexneri*; 16, *Shigella sonnei*; 17, *Spirosoma radiotolerans*; 18, *Streptococcus thermophilus*; 19, *Thermus thermophilus*; 20, *Vibrio cholera* O1; 21, *Yersinia enterocolitica*.

**TABLE 1** Test/train data set summary

| Set | Total | Tolerant | Radiosensitive |
|---|---|---|---|
| Train | 40 | 30 | 10 |
| Test | 21 | 17 | 4 |

final model to determine which domains were most important for the model's ability to correctly classify radiation tolerance. Further details about predictor selection are provided in Materials and Methods.

## Model training and validation

The final model was built using the RandomForest function of the R package caret (38) via a 10-cross-validation on the train set. The accuracy of the final model on the train set was 0.875. The final model was unable to correctly classify 5 out of 40 (12.5%) of the bacteria in the train set, making three false negative and two false positive errors (Fig. 1C; Table S3). Encouragingly, these misclassifications were of species with $D_{10}$ values within 100 Gy of the classification cut-off, suggesting species classified as radiosensitive are still likely to have relatively low $D_{10}$ values. When the classifier was applied to the species of the test set that were hidden from the model during training, the accuracy was 0.900. As with the train set, the two misclassifications were of species with experimentally determined $D_{10}$ close to the model classification cutoff (Fig. 1D; Table S3).

## Using TolRad to screen species of the human microbiome reveals the radiosensitivity of the *Bacteroidota* phylum

To discover novel radiosensitive bacteria within the HMB, we applied TolRad to the proteomes of 152 bacterial strains that had previously been detected within samples originating from the human microbiome. This data set included 37 species of the official Human Microbiome Project strain collection hosted by ATCC (Tables S2 and S4) (39), as well as 28 species identified from the human skin microbiome (40, 41), 46 species from the human oral cavity microbiome (42), and 41 species from the human gut microbiome (43). For this analysis, as for the construction of TolRad, we used the Pfam domain annotations hosted on UniProt for each species, selecting species with ATCC strain ID to support downstream experimental validation. In total, we identified 34 putative radiosensitive species (Table 3; Table S5), of which 10 belonged to the ATCC-hosted NIH Human Microbiome Project. A secondary literature search for $D_{10}$ values associated with these 34 putative radiosensitive species uncovered support for the radiosensitive classification of *Klebsiella pneumoniae* (44). All UniProt proteome classification predictions are presented in Table S4.

We examined the phylum level diversity of the bacteria characterized as radiosensitive from the HMB set compared with that of train/test. While the train/test set

**TABLE 2** TolRad predictors

| ID | Pfam short name | Mean decrease in accuracy | Mean occurrence per proteome | Example from *E. coli* and Panther function |
|---|---|---|---|---|
| PF00300 | Pyridine nucleotide-disulfide oxidoreductase | 7.58 | 15.52 | norW: nitric oxide reductase<br>trxB: thioredoxin reductase<br>ndh: type II NADH:quinone oxidoreductase |
| PF07992 | Histidine phosphatase | 11.57 | 6.87 | ais: lipopolysaccharide metabolic process<br>gpmA/B: glycolytic process<br>sixA: signal transduction |
| PF03466 | LysR substrate binding domain | 8.95 | 63.42 | cysB: TF regulator of response to X-rays<br>oxyR: ROS-responsive TF<br>lysR: TF lysine biosynthesis |
| PF13411 | HTH family regulatory protein | 5.5 | 7.36 | cueR: copper-responsive TF<br>zntR: zinc-responsive TF<br>mlrA: regulator of biofilm formation |

**TABLE 3** UniProt species summary

| Set | Species | Phyla | Radiosensitive (predicted) |
|---|---|---|---|
| NCBI/ATCC collection | 37 | 5 | 10 |
| Human skin | 28 | 4 | 4 |
| Human oral cavity | 46 | 6 | 9 |
| Human gut | 41 | 6 | 11 |
| Total | 152 | | 34 |

only contained radiosensitive bacteria from the *Proteobacteria* phylum (Fig. 2A), TolRad predicted radiosensitive species from the HMB within the *Actinobacteria*, *Firmicutes*, and *Bacteroidetes* phyla (Fig. 2A). Of particular interest was the prediction that 19 of the 29 *Bacteroidetes* species were characterized as radiosensitive (Table S5). Several species of this phylum are highly abundant within the human gut microbiome, including *Bacteroides thetaiotaomicron* (*B. thetaiotaomicron*) (45). To validate the ability of TolRad to predict the IR tolerance of bacteria beyond the phyla represented as radiosensitive ($D_{10} < 200$ Gy) in the train/test set, we experimentally determined the tolerance of *B. thetaiotaomicron* for IR. Using an X-ray source and the CFU assay, we calculate the $D_{10}$ of *B. thetaiotaomicron* to be 110 Gy (Fig. 2B). As the train/test set contained both radiosensitive and IR-tolerant species from the *Proteobacteria* phylum, we experimentally validated the $D_{10}$ values of *Acinetobacter baumannii*, a *Proteobacteria* species predicted to be IR-tolerant, and of *Pseudomonas aeruginosa*, a *Proteobacteria* species predicted to be radiosensitive. Both classifications were experimentally corroborated, with the $D_{10}$ of *Acinetobacter baumannii* determined to be 400 Gy (Fig. 2C) and the $D_{10}$ of *Pseudomonas aeruginosa* determined to be 130 Gy (Fig. 2D). In summary, TolRad identified 34 putative radiosensitive bacteria, including 10 from the ATCC Human Microbiome Project strain collection (39). Of these 34, 1 had previously been investigated for IR tolerance and was found to have $D_{10}$ values in line with the classification as radiosensitive, and 2, including 1 from a phylum without radiosensitive examples in the train/test, were experimentally validated to have $D_{10}$ values below the 200 Gy classification cutoff (Fig. 2B). Additionally, we validated the ability of TolRad to correctly differentiate radiosensitive and tolerant species within the *Proteobacteria* phylum. Through this finding, we demonstrate that TolRad can be applied to bacterial genomes to which Pfam domains have already been assigned, including the upward of 46,000 bacterial proteomes currently on UniProt (https://www.uniprot.org/).

## TolRad remains accurate when using *de novo* Pfam annotations assigned using EggNOG

We next sought to expand the utility of TolRad beyond pre-annotated UniProt proteomes to MAGs. Since MAGs are constructed from environmental samples, they are unlikely to match with existing annotated genomes. For this reason, MAGs need to undergo both gene calling and Pfam annotation before TolRad can be applied. Additionally, MAGs, unlike UniProt proteomes, are often only partial genome assemblies (46).

To test the consistency of classifications made by TolRad on *de novo* Pfam annotations, the genome annotation pipeline EggNOG-Mapper (32) was used to assign coding regions and annotate Pfam domains directly from the genome assemblies of the train/test set. The workflow we used for processing and classifying MAGs is described in Fig. 3A. TolRad returned the correct classification for 59 of the 61 (accuracy of 0.967%) EggNOG-Mapper annotated genomes in the train/test set (Fig. 3B; Table S3).

To test the ability of TolRad to handle incomplete genomes, a common occurrence in MAG data sets (46), we generated mock incomplete MAGs that ranged from 90% to 50% completion, in increments of 10%. This was done by randomly sampling the Pfam domains generated from each EggNOG-Mapper-produced annotation file. After mock degradation, the relative frequency of the Pfams used by TolRad was recalculated, and

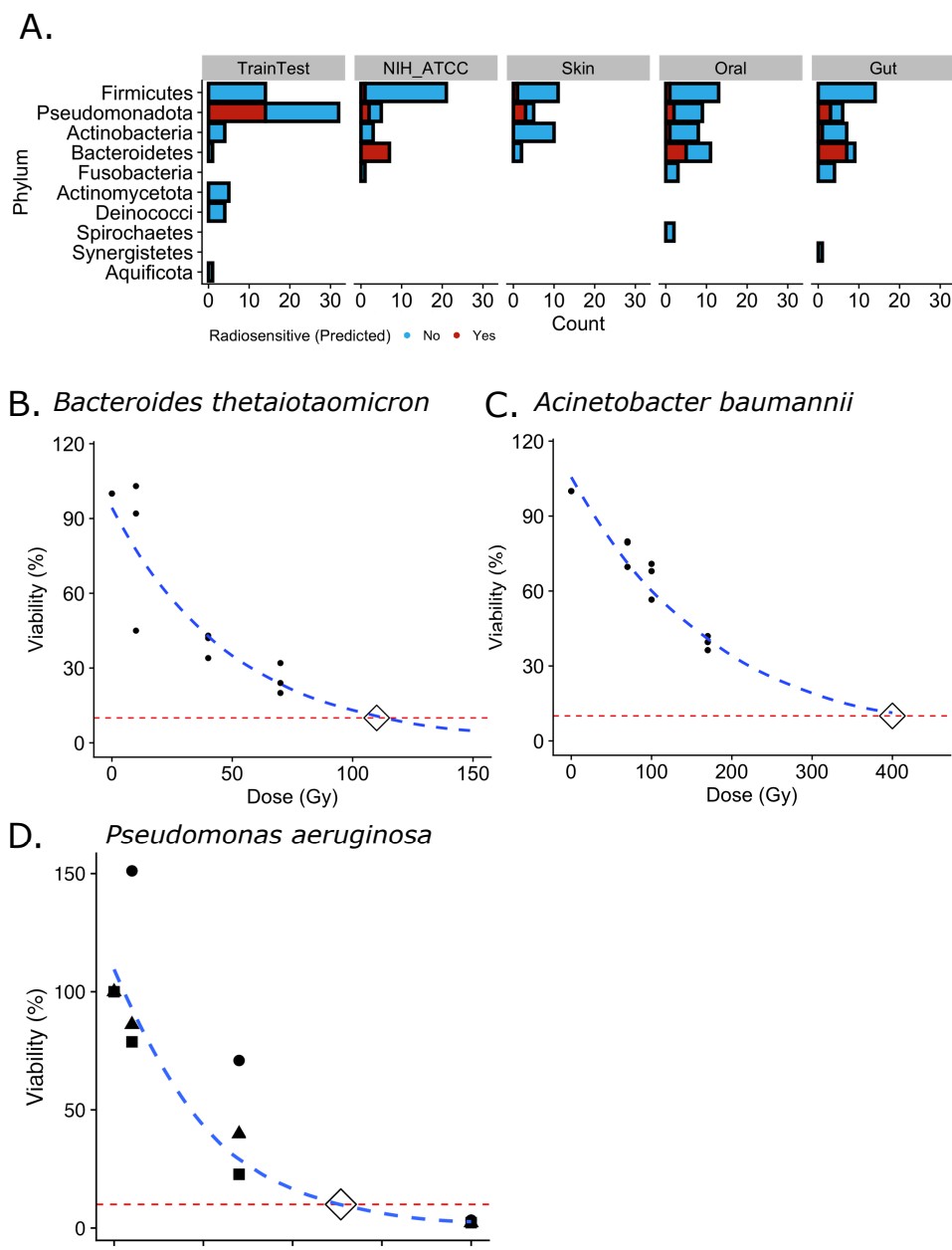

**FIG 2** Application of TolRad to bacteria isolated from the human microbiome. (A) Counts of bacteria species by phyla. Phylum diversity and tolerance predictions (by color) within the train/test set, the ATCC-hosted NIH Human Microbiome Project, and additional species isolated from the human skin, oral, and gut microbiome. (B) Survival, determined via CFU, of *Bacteroides thetaiotaomicron* at 10, 40, and 70 Gy. From three technical replicates, the $D_{10}$ was determined to be 110 Gy. Survival at 200 Gy is predicted to be 1.8%. (C) Survival, determined via CFU, of *Acinetobacter baumannii* at 10, 70, and 170 Gy. The $D_{10}$ was determined to be 400 Gy. (D) Survival, determined via CFU, of *Pseudomonas aeruginosa at* 10, 70, and 200 Gy. The $D_{10}$ was determined to be 130 Gy.

the tolerance for IR was reclassified by TolRad. The percent of the mock incomplete MAGs correctly classified at each level of completeness were recorded. As shown in Fig. 3C, the ability of TolRad to classify bacterial tolerance to IR worsened as the completeness of the mock MAGs decreased in a linear fashion; however, the classification rate stayed above 85% correct until 40% of the annotations had been removed. We also examined

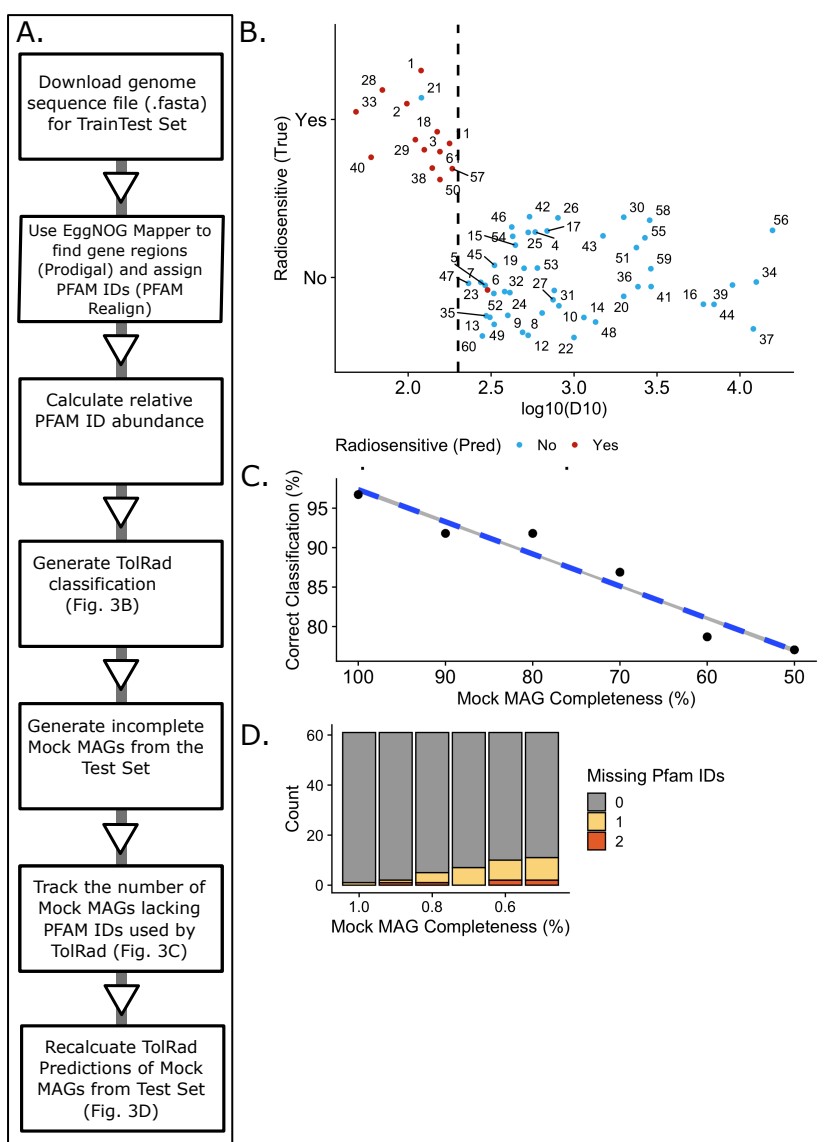

**FIG 3** (A) Workflow for generating TolRad predictions on *de novo* genome annotations. First, the genome sequence is downloaded. EggNOG-Mapper is used to assign Pfams. Pfam abundance is calculated. TolRad classifications are made. Then, Pfam domains were randomly removed, and the classification was determined again. (B) *De novo* genome annotations were generated for the 61 species of the train/test set. Species numbers (graph ID) are in Table S3. These "mock MAGS" were classified using TolRad. Color denotes the classification assigned by TolRad. Red denotes a species classified as radiosensitive. Blue denotes a species classified as tolerant. The dashed line at 200 Gy is the cutoff used for defining radiosensitive vs tolerant species. (C) The mock MAGs were randomly degraded to 90%, 80%, 70%, 60%, and 50%. The Pfam abundances were recalculated along with the TolRad classification. The percent of correct classifications at each level is reported. (D) The number of missing predictors at each level of degradation within the mock MAGs from (C).

the relationship between mock incomplete MAGs and the rate at which predictor Pfam domains were lost (Fig. 3D); however, even at 50% incomplete, less than 20% of the mock incomplete MAGs were lacking even one of the four Pfam domains used by TolRad. In summary, the use of Pfam domain frequency determined using the EggNOG-Mapper 5.0 (32) genome annotation pipeline did not decrease the ability of TolRad to correctly classify bacterial tolerance for IR, and TolRad performance was reasonably robust on partial genomes.

**TABLE 4** MAG summary

| Data set | Total MAGs | Source | High-quality MAGs | Phyla | Classification | |
|---|---|---|---|---|---|---|
| | | | | | Radiosensitive | Tolerant |
| CHA | 31 | (34) | 26 | 5 | 2 | 24 |
| Deep ocean | 317 | (35) | 250 | 21 | 78 | 172 |
| HMB | 96 | (47) | 91 | 4 | 7 | 84 |
| Total | 444 | | 367 | | 87 | |

## Applying TolRad to a collection of MAGs identified the deep sea and human gut as harboring a number of putative radiosensitive bacteria

To demonstrate the utility of TolRad for identifying radiosensitive species from within environmental samples and gain insights into the ecological distribution of radiosensitive bacteria, we applied TolRad to three collections of previously assembled MAGs collected from diverse environments.

First, we examined a collection of MAGs originating from the human microbiome, representing the microbiome of the skin (HSM) and the gut (HGM) (47). The 91 high-quality MAGs (completeness >60%) in this data set had been previously assigned to four phyla (*Firmicutes*, *Proteobacteria*, *Actinobacteriota*, and *Bacteroidota*) and had a mean completeness of 93.25%. Within this data set, only three MAGs lacked one of the Pfam domains used by the model. TolRad characterized 7 of the 91 MAGs (7.69%) as radiosensitive (Table 4; Table S4). All putative radiosensitive MAGs were members of either *Bacteroidota* or *Firmicutes* phylum (Fig. S2A). This finding agreed with the radiosensitive predictions made on the species of the HMB UniProt data set (Fig. 2A). Interestingly, all the MAGs predicted to be radiosensitive were collected from the HGM (17 out of 64, 26.56%), and no radiosensitive bacteria were predicted from the HSM-collected MAGs (Fig. 4A). In the authors' original analysis, the 91 MAGs were binned to 46 National Center for Biotechnology Information (NCBI) taxonomies, and 18 of these NCBI taxonomies included multiple MAGs. We expected that MAGs within the same NCBI taxonomy would have the same tolerance for IR and found that TolRad had consistent tolerance predictions for 17 of 18 NCBI taxonomies (Fig. S2B).

Next, we examined a set of MAGs collected from seasonal glacial surface streams in the CHA (34). This MAG set represents an environment where, due to the high levels of solar radiation, we did not expect to identify radiosensitive bacteria (48). The High Arctic MAGs had a lower mean completeness (77.15%) compared with the other MAG sets, although the number of MAGs lacking the predictor Pfam domains was minimal. As we expected, of the 26 MAGs with high-quality MAGs, only 2 (7.6%) were predicted to be radiosensitive, and both were identified as *Proteobacteria* (Fig. S2C; Table S4).

To discover a greater diversity of radiosensitive bacteria, we applied TolRad to a set of 312 MAGs collected from the deep ocean (35). The original publication assigned these MAGs to 26 phyla, and the MAGs had a mean completeness of 84.2%. Due to the total lack of UV radiation in the deep sea, and the suspected role of ROS, such as UV exposure, in the evolution of IR tolerance (25), we expected to identify a diversity of radiosensitive bacteria in this MAG data set. Compared with the HMB data set and the mock incomplete MAGs, the deep ocean collection had a greater number of MAGs that were missing predictor Pfam domains (Table S3). This finding may be due to the diversity of the deep ocean collection, which included many phyla not represented in the train set (Fig. 4B). We observed that MAGs assigned to phyla not in the train/test set had a greater rate of MAGs that lacked the Pfam domains used by TolRad than the MAGs assigned to phyla in the train/test set. Because we were unable to model the impact of missing predictor Pfam domains on TolRad accuracy, MAGs that were lacking two or more Pfam domains were excluded from classification. This left 250 MAGs from 21 phyla. Of the MAGs examined, 78 were predicted to be radiosensitive (31.2%) (Fig. 4A; Table S4), the highest percent of the MAG collections we examined. These MAGs came from 17 phyla, with many MAGs belonging to the *Acidobacteriota*, *Planctomycetota,* or *Marinisomatota* phyla (Fig. 4B),

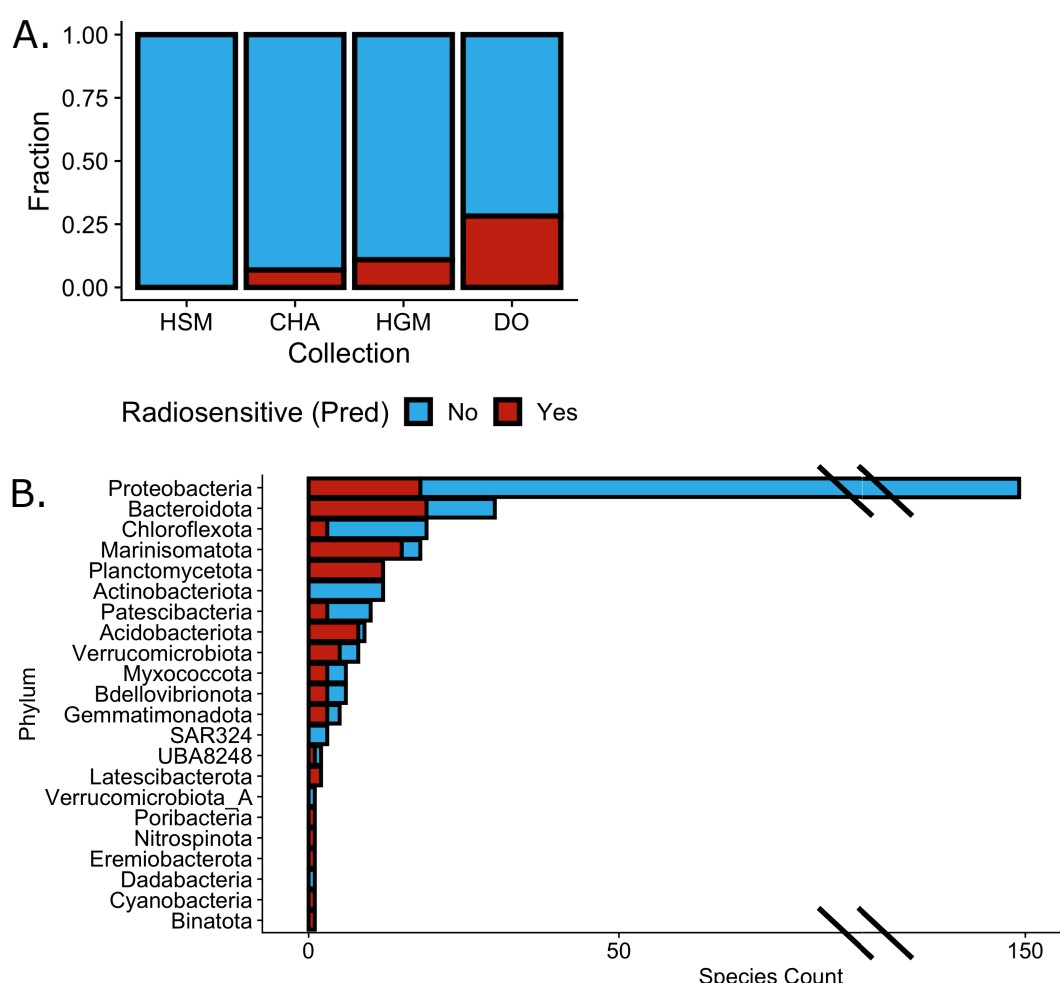

**FIG 4** Application of TolRad to three MAG collections. (A) As described in Fig. 3, *de novo* annotations were assigned to previously published environmentally collected MAGs. The fraction of previously published MAGs classified as radiosensitive (red) or tolerant (blue) across MAGs collected from the HSM, a glacial stream in the CHA, the HGM, and the water column of the deep ocean (DO). (B) Counts of MAGs, by phylum, from the deep ocean collection that were classified as radiosensitive or tolerant.

suggesting that species from these phyla have potential for further investigations of the mechanism of IR sensitivity and studies of IR biomarkers.

## DISCUSSION

TolRad is a random forest binary classifier that uses protein annotations, which can be generated directly from genome assemblies, to predict if a bacterium is radiosensitive ($D_{10} < 200$ Gy). TolRad is available as a stand-alone R package and can be accessed at https://github.com/philipjsweet/TolRad and can be applied to both reference proteomes and MAGs. The ability of TolRad to identify radiosensitive bacteria species on which it had not been trained on was demonstrated using a test set of bacteria (with experimentally determined $D_{10}$ values) that were excluded from the

**TABLE 5** Strains

| Species | Strain designation | ATCC ID |
|---|---|---|
| *Bacteroides thetaiotaomicron* | VPI 5482 | ATCC 29418 |
| *Pseudomonas aeruginosa* | PA-103 | 29260 |
| *Acinetobacter baumannii* | 2208 | ATCC 19606 |

construction of TolRad (Fig. 1D). The change in accuracy from the train set to the test set was negligible (from 0.875 to 0.900). The ability of TolRad to make radiosensitive identifications for species from phylum without radiosensitive representation in the train/test set was demonstrated by experimentally validating the radiosensitive classification of a *Bacteroidetes* species and tolerant classification of an *Acinetobacter* species (Fig. 2B). The generalizability of TolRad was shown by applying TolRad to both pre-annotated proteomes from UniProt and *de novo*-assembled proteomes to identify putative radiosensitive species from across 19 phyla. Additionally, TolRad can be used to understand the general IR tolerance of bacterial communities (Fig. 4A). We also demonstrate that TolRad can be used to classify the tolerance of bacteria that have been well studied, such as those from the ATCC Human Microbiome Collection (39) (Fig. 2), as well as MAGs, such as those previously assembled from the deep sea (34) (Fig. 4B). Applying TolRad has allowed us to greatly expand the number and diversity of bacteria that are likely to be sensitive to IR exposure. We have uploaded TolRad to GitHub as an R script (https://github.com/philipjsweet/TolRad), allowing for those to computationally screen collections of bacterial genomes and for the integration of TolRad into metagenomic analysis pipelines. In summary, we have created a predictive model of bacterial tolerance for IR that relies exclusively on genomic annotations and demonstrated that this model can be widely deployed.

## Insights into the genetic traits of radiosensitivity

In addition to the construction of TolRad, this study also allows for a greater understanding of the bacterial traits that are associated with radiation tolerance. Previous work by the Daly Lab (23, 25, 26) has demonstrated that the intracellular ratio of manganese to iron (Mn/Fe) is a predictor of a species' tolerance for IR. This correlation between radiation tolerance and the intracellular ratio of Mn/Fe was also observed in UV-C-tolerant bacteria (49). Additionally, studies have identified proteins correlated with resistance to IR (50, (51). While genetic explanations have been offered for the sensitivity of specific bacteria, such as a large number of proteins with heme (i.e., iron-binding) domains in the radiosensitive bacteria *S. oneidensis* (26, 52), a broadly applicable understanding of the genetic origins of radiation sensitivity has not been proposed. The construction of TolRad required the identification of Pfam domains informative of radiation tolerance; we present these findings, as well as the mean decrease in accuracy and examples of *Escherichia coli* genes with these domains, in Table 2. We found that PF07992 *pyridine nucleotide-disulfide oxidoreductase* domains are often found in reductases within known roles regulating intracellular ROS such as *norW, txrB,* and *ndh*. The importance of PF07992 is in agreement with previous work demonstrating the importance of limiting ROS spread to ensure IR survival (26). PF00300 *histidine phosphatase superfamily* (*branch 1*) was the second most important predictor in the model. PF00300 domains are found in a diverse set of proteins (Table 2), complicating the interpretation of its contribution to IR tolerance. The contribution of the PF03466, *LysR substrate binding domain*, is often found in ligand response transcription factors, and it has been previously noted that while the radiosensitive bacterium *S. oneidensis* has 52 proteins in the LysR family, the radioresistant bacterium *D. radiodurans* only has two (53). Similarly, the connection of PF00849 *RNA pseudouridylate synthase*, a domain found only in RNA pseudouridylate synthase, to the classification of bacterial IR tolerance is unclear. A complication of interpreting random forest predictors is that the random forest model does not produce an equation, with coefficients that speak to the relationship between the predictor and the response variable. Interestingly, none of the Pfam domains utilized by TolRad to predict IR tolerance involve iron-binding domains or metal ion importer domains, as may have been expected given the correlation between IR tolerance and the intracellular ratio of Mn/Fe. Future examinations of specific radiosensitive bacteria, such as those identified in this study, and focused studies into the proteins with the Pfam domains that we have correlated with IR tolerance will be required to fully understand the biological implications of these four Pfam domains for IR tolerance.

## Applications of TolRad

There are several applications for TolRad, including providing context for metagenomic studies of bacterial communities after IR exposure (19), searching for bacterial sources of low-dose biomarkers (52), and guiding the selection against radiosensitive species for bioremediation development (17). Currently, the only way to determine the sensitivity of a bacteria species to IR is to conduct extensive experimentation, requiring both the ability to culture the bacteria of interest and a high-powered source of IR. While survival screens have enabled the isolation of bacteria with extreme tolerance for IR (30, 54, 55), no such discovery studies have been conducted for bacteria sensitive to IR. As an example of applying TolRad to screen existing proteomes, we applied TolRad to a collection of 152 proteomes downloaded from UniProt, including the ATCC human microbiome strain collection (Table S4). TolRad identified 34 putative radiosensitive species ($D_{10} < 200$ Gy), including several of the most abundant bacteria in the human gut (43). We then demonstrated experimentally that *B. thetaiotaomicron* is in fact radiosensitive, with a $D_{10}$ of 110 Gy. Excitingly, this prediction was correct, despite being made on species from phylum on which TolRad was not trained (Fig. 2C). Additionally, TolRad correctly differentiated radiosensitive from tolerant species within the *Proteobacteria* phylum.

As another example of applying TolRad for screening bacterial communities for radiosensitive species, we applied TolRad to MAGs to a collection of deep sea MAGs and identified 78 candidate species from 17 phyla as radiosensitive (Fig. 4B). TolRad will allow other researchers to scan bacterial genomes rapidly and determine which species from a population is radiosensitive. These predictions will help guide future exploration of bacterial tolerance of IR.

## Taxonomy of radiosensitive species

We were also interested in the taxonomy of radiation tolerance, as when we began this study, bacteria with a low survival threshold for IR exposure had only been identified within the *Proteobacterium* phylum. To our knowledge, the data set that we collected for TolRad is the most comprehensive set of bacterial acute $D_{10}$ values published to date. We observed, within the train/test set, that sensitivity for IR was limited to *Proteobacteria*; however, when we expanded our search to bacteria of unknown IR tolerances using TolRad, we found a diversity of putative radiosensitive species. Across the UniProt proteomes and the three collections of MAGs, we applied TolRad to over 500 proteomes, representing 44 phyla, and identified 121 putative radiosensitive species across 21 phyla (Table S3). Since there is a minimal amount of naturally occurring IR on Earth, previous studies have suggested extreme IR tolerance, observed in multiple phyla (Fig. 1B; Table S1), may have evolved as a response to bacteria living in environments with elevated ROS (31). Previous work with marine bacteria noted that species collected from the subsurface had a lower tolerance for UV radiation and hydrogen peroxide, both ROS-inducing, than those collected at the ocean surface. The authors suggest that this may be due to a lower selective pressure from sunlight exposure on the subsurface species (48). This dynamic could also explain the difference in radiation tolerance between species of the HSM and the HGM (Fig. 4A). This idea is supported by the low number of radiosensitive bacteria that TolRad classified from the high UV (CHA) and high hydrogen peroxide (HSM) MAGs. Based on the predictions made by TolRad, we suggest that, in a similar way, IR sensitivity could be prevalent in UV-sheltered environments, such as the deep sea (Fig. 4A) and the human gut (Fig. 4A). Identifying environments that are rich in IR-sensitive species can aid in understanding the common mechanistic traits of IR-sensitive species.

## Limitations

The greatest limitation of TolRad is the size of the test/train set. We were only able to collect $D_{10}$ values from control (exposures conducted at room temperature and in growth media or PBS) conditions for 61 bacteria (Table S1), Additionally, there are

multiple laboratory sources of IR (i.e., X-ray, Co60, and C137), which are regularly grouped, despite having variable effects on biological systems. The train/test Set only represented seven bacterial phyla, and radiosensitive species were only found within one of those phyla. The train/test set was heavily skewed toward *Proteobacteria* (41.3%) and *Firmicutes* (30.0%), so it is possible that the model is more accurate on these phyla than on others. The experimental validation that we conducted on a species of the *Bacteroidetes phylum* supports the use of TolRad beyond the phyla of the train/test set; however, similar initial validation experiments would be prudent for radiosensitive predictions made on species from additional phyla. To prevent overfitting to rare Pfam domains, only Pfam domains present in the proteomes of all the species within the test set were used, and this selection criteria worked well when TolRad was applied to species from HMB and the High Canadian Artic, all of which had a similar range of phyla as the train set. When examining the deep ocean MAG collection, we noted that some of the phyla had a much higher rate of proteomes that were missing predictor Pfam domains (Table S4). For this reason, TolRad reports the number of missing predictor Pfam domains for each genome that is classified, and in this paper, we only discuss classifications of putative radiosensitive species with at least three of the four predictors. Undoubtedly, TolRad will make misclassifications; however, the ease of incorporating additional experimentally determined $D_{10}$ values into the train set used to build TolRad means that future findings could be incorporated into TolRad to bolster the predictive power of this tool.

In summary, in this paper, we have described TolRad, a strictly computationally based classifier of bacterial tolerance for IR. We have demonstrated the accuracy of TolRad beyond the bacteria species on which it was trained. We further demonstrate the ability of TolRad to identify putative novel bacteria with a low survival threshold for IR exposure. Additionally, we present the first experimental characterization of the IR sensitivity of *B. thetaiotaomicron*, an abundant member of the human microbiome. Further collection and validation of radiation phenotypic straits in bacteria should provide additional data upon which to continue to improve TolRad and other similarly trained models.

## MATERIALS AND METHODS

### Collection of $D_{10}$ values

The species used in the train/test set are presented in Tables S1 and S2. Only $D_{10}$ values determined using an acute dose from a source of ionizing radiation, at room temperature, and in a neutral liquid matrix (i.e., growth media, PBS, or water) were considered for inclusion in the train/test set. For species with multiple reported $D_{10}$ values, the mean was used to assign a tolerance classification (Table S2). The UniProt proteome from which Pfam domains were obtained is also provided. The lowest 20% of the $D_{10}$ values were classified as radiosensitive. Bacteria were classified as radiosensitive if the mean $D_{10}$ was below 200 Gy and tolerant if the mean $D_{10}$ was above 200 Gy.

### Selection of Pfam domains and model construction

The Pfam domains for each bacteria in the train/test set were acquired from UniProt. When possible, the "reference" version of the proteome was used. Otherwise, the most complete proteome was used. No proteomes classified by UniProt as "low coverage" were used in the train/test set. The UniProt proteome IDs are reported in Table S1. The train/test set was randomly split into a train set ($n = 40$) and a test set ($n = 21$). The train set was used for the construction of TolRad. The raw counts of each Pfam domain were calculated at the species level and across the train set. As shown in Fig. 1B, we started with 7,409 unique Pfam domains. We then selected domains that were universal to the train set, which left 395. Of these, we selected those with a mean occurrence per species greater than 2; this left 132. For each species, the relative frequency of each Pfam domain, against the total number of Pfam domains within each species, was calculated. The Boruta (37) feature selection algorithm (maxRuns = 500) was used to

select seven Pfam domains for which the relative frequency (out of the total number of Pfam domains for the species) was correlated with the species IR tolerance classification (radiosensitive or tolerant). The relative frequency of these domains was used to construct a random forest model using the RandomForest algorithm of the caret R library (https://cran.r-project.org/web/packages/caret/index.html) to classify bacteria as tolerant of IR or radiosensitive. A 10× cross-validation was used to train the model. To prevent overfitting the model to the species of the train set, the mean decrease in accuracy for each of the seven predictor Pfam domains was determined, and predictors were removed stepwise, starting with the domain with the lowest mean decrease in accuracy. Of the seven domains, four were determined to be required for correct classification and were retained for the final model (Table 2).

## Making predictions using TolRad UniProt

For the classification of HMB UniProt species, Pfam annotations were downloaded from UniProt (www.uniprot.org) in August of 2022. In Table S4, all of the species on which TolRad radiation tolerance predictions were made can be found. For each entry, where applicable, the following is provided: UniProt species names, ATCC strain ID (ATCC_Strain), ATCC database name (ATCC_Name), internal reference ID (ID), total number of missing predictor Pfam IDs (missing Pfams) and taxonomy (phylum) as well as the literature source suggesting that the species is in the HMB (source), and the MAG ID. The random forest model trained and validated in this paper is the basis for the TolRad R package, which has been uploaded to GitHub (https://github.com/philipjsweet/TolRad) along with a user guide. Briefly, the user provides a path to a folder with a collection of Pfam genome annotations of bacterial species for which radiation tolerance classifications are desired and the function outputs a data frame with the classification of each species.

## Processing MAGs

The train/test genome assemblies associated with the UniProt proteomes used to train and test TolRad were downloaded from EMBL-EBI (https://www.ebi.ac.uk/genomes/). The assembly IDs are in Table S4. MAGs from the CHA (34) were downloaded, along with taxonomy assignments and completeness scores from https://figshare.com/articles/dataset/Borup_Fiord_Pass_-_Metagenome_Assembled_Genomes_MAGs_/9767564. Human microbiome (47) samples were downloaded from the Sequence Read Archive (https://www.ncbi.nlm.nih.gov/bioproject/), and taxonomy assignments and completeness scores were acquired from the supplemental figures. The genome assemblies test/train set and the MAGs HMB and the CHA were processed using EggNOG-Mapper (version 5.0) (32) (--pfam_realign realign --itype genome --genepred prodigal). The deep sea MAGs (35) were downloaded, with Pfam annotations, from https://malaspina-public.gitlab.io/malaspina-deep-ocean-microbiome/. Pfam domain frequency and TolRad predictions were generated as described above.

## Bacterial strains and culture conditions

All strains used to test TolRad tolerance predictions are provided in Table 5. *Bacteroides thetaiotaomicron VPI 5482* (56) was in grown brain heart infusion media (BHIM) (Fisher Scientific) with anaerobic conditions without shaking at 37°C. For all experiments, unless otherwise stated, cells were grown from single colonies in 10 mL of BHIM in Hungate tubes. On the morning of exposure, 1.0 mL of O/N culture was diluted into 9.0 mL of fresh media to an optical density ($OD_{600}$) of ~0.1 in a Hungate Tube. Cells were grown for ~5 h until an $OD_{600}$ of ~0.4. 5 mL of culture was sealed in de-gassed Nasco Whirl-Pak bags, sealed in anaerobic growth bags, and laid flat on the exposure tray to ensure even dosage across the sample.

 *Acinetobacter baumannii* 2208 (57) and *Pseudomonas aeruginosa* PA103 (58) were grown in Luria broth (LB) (Fisher Scientific) at 37°C with shaking. For all experiments,

unless otherwise stated, cells were grown from single colonies in 5 mL of LB and grown on LB plates. For exposures, cultures were grown to an $OD_{600}$ of 0.6 before being split into 5 mL aliquots and sealed in Nasco Whirl-Pak bags for exposure.

## X-ray exposures

When exposing cells to IR, a Faxitron 225 MultiRad X-ray irradiator was used. The machine was set to 12 mA, 220 kV with a shelf height of 44.5 cm for doses below 10 Gy, at a height of 37 cm for doses between 10 and 70 Gy and a height of 29.5 cm for doses above 70 Gy. A 0.5 mm aluminum filter ensured that only high-energy X-rays were delivered. Exposures were conducted at room temperature (24°C).

## CFU assays

For survival assays, three biological replicates were grown to and exposed as described above. After exposure, cells were serially diluted in PBS and spread on BHIM or LB plates using beads. Three technical replicates were conducted per biological replicate. CFUs were counted, and the mean sham (0 Gy)-exposed plate count of the three technical replicates was used as the baseline against which the technical replicates of the doses of that biological replicate were compared to determine the surviving fraction. These data were fitted to a logistical regression model to calculate the $D_{10}$ value.

## ACKNOWLEDGMENTS

We would like to thank Daryl Barth (University of Texas at Austin) for assistance in running the EggNog-Mapper annotations.

This research was funded by a grant (FA9550-20-1-0131) from the Air Force Office of Scientific Research as well as a grant (HDTRA1-17-1-0025) from the Defense Threat Reduction Agency. This work was also supported by a grant (W911NF22S0002) from the Intelligence Advanced Research Projects Activity.

## AUTHOR AFFILIATION

[1]McKetta Department of Chemical Engineering, University of Texas at Austin, Austin, Texas, USA

## AUTHOR ORCIDs

Philip Sweet http://orcid.org/0000-0002-0802-693X
Lydia Contreras http://orcid.org/0000-0001-5010-5511

## FUNDING

| Funder | Grant(s) | Author(s) |
| --- | --- | --- |
| DOD | Defense Threat Reduction Agency (DTRA) | HDTRA1-17-1-0025 | Lydia Contreras |
| DOD | USAF | AMC | Air Force Office of Scientific Research (AFOSR) | FA9550-20-1-0131 | Lydia Contreras |
| DNI | Intelligence Advanced Research Projects Activity (IARPA) | W911NF22S0002 | Lydia Contreras |

## AUTHOR CONTRIBUTIONS

Philip Sweet, Conceptualization, Data curation, Formal analysis, Investigation, Methodology, Validation, Visualization, Writing – original draft, Writing – review and editing | Lydia Contreras, Conceptualization, Funding acquisition, Investigation, Project administration, Supervision, Writing – review and editing.

## ADDITIONAL FILES

The following material is available online.

### Supplemental Material

**Supplemental Figure 1 (Spectrum03838-23-s0001.tiff).** Traits of the Test/Train set.
**Supplemental Figure 2 (Spectrum03838-23-s0002.tiff).** Taxonomy of MAGS.
**Supplemental Tables 1-5 (Spectrum03838-23-s0003.xlsx).** Tables of additional data.

### Open Peer Review

**PEER REVIEW HISTORY (review-history.pdf).** An accounting of the reviewer comments and feedback.

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
