## [Reviewer comments · Microbiology Spectrum]

Microbiology Spectrum

TolRad: A model for predicting radiation tolerance using Pfam annotations identifies novel radiosensitive bacterial species from reference genomes and MAGs

Philip Sweet, Matthew Burroughs, Sungyeon Jang, and Lydia Contreras

Corresponding Author(s): Lydia Contreras, The University of Texas at Austin

Review Timeline:

Submission Date:	November 2, 2023
Editorial Decision:	April 17, 2024
Revision Received:	June 4, 2024
Accepted:	June 20, 2024

Editor: Jonathan Jacobs

Reviewer(s): Disclosure of reviewer identity is with reference to reviewer comments included in decision letter(s). The following individuals involved in review of your submission have agreed to reveal their identity: Paul Jaak Janssen (Reviewer #1)

Transaction Report:

DOI: <https://doi.org/10.1128/spectrum.03838-23>

Re: Spectrum03838-23 (ToIRad: A model for predicting radiation tolerance using Pfam annotations identifies novel radiosensitive bacterial species from reference genomes and MAGs)

Dear Dr. Lydia M Contreras:

Thank you for the privilege of reviewing your work, and for your patience while I secured comments from reviewers with the appropriate experience. Below you will find my comments, instructions from the Spectrum editorial office, and the reviewer comments.

Overall, I thought this was an excellent body of work that was communicated well by the manuscript you submitted. The ToIRad application presents some interesting possibilities for large scale screening of both MAG and isolate assembly databases. A few minor issues I found need to be corrected: 1/ in Table 5, the "ATCC ID" is not correct for either of the strains listed, but instead you have the strain designation listed. I would suggest listing both, first by changing the header of the table to "strain designation" for the existing information, and then adding a third column for ATCC ID (which would be ATCC 29418 and ATCC 19606 for the respective strains of *B. thetaiotaomicron* and *A. baumannii* listed). 2/ The figure axis labels in Fig 2A appear to have been swapped (phylum/count).

In addition to my two minor corrections, please review and respond to all of the comments from the reviewers. Where possible, please make the appropriate changes to the manuscript to meet their recommendations. Please note, reviewer#1's comments may be attached as a word document as opposed to inline below.

After you have made these adjustments, please return the manuscript within 60 days; if you cannot complete the modification within this time period, please contact me. If you do not wish to modify the manuscript and prefer to submit it to another journal, notify me immediately so that the manuscript may be formally withdrawn from consideration by Spectrum.

Revision Guidelines

Sincerely,
Jonathan Jacobs
Editor
Microbiology Spectrum

Reviewer #1 (Public repository details (Required)):

it may be helpful to others to provide the Train/Test set

Reviewer #1 (Comments for the Author):

see my report attached

Reviewer #2 (Comments for the Author):

In this well-written paper, the authors reported a computational model using de novo generated genome annotations to classify a bacterium as tolerant of IR or as radiosensitive, which allowed for rapid screening of bacterial communities for low tolerance species that are of interest for both mechanistic studies into bacterial sensitivity to IR and biomarkers of IR exposure. Overall, this is a useful tool with very high prediction accuracy for microbiologists and radiologists. The reviewer suggests the publication of this paper after minor modifications. The authors briefly mentioned that Mn/Fe had been used as a predictor of a species' tolerance for IR. Based on their big data analysis, the authors should give more discussions regarding the genetic and biomedical mechanisms for bacterial IR tolerance.

TolRad: A model for predicting radiation tolerance using Pfam annotations identifies novel radiosensitive bacterial species from reference genomes and MAGs

Comments and Suggestions for the Author:

General (for text including legends):

- the abstract lacks emphasis on the importance/relevance of predicting radiosensitivity; I am thinking in terms of gut microbial dysbiosis during pelvic radiotherapy or changes in human microbiomes under harsh space conditions, or the deployability of bacteria in radiation-intensive or radioactively-contaminated environments.
- make sure all hyperlinks are active i.e. point to functional webpages
- use SI units only i.e. mL, not ml
- always use a space between a given value and a unit i.e. 5 Gy, not 5Gy, or 5 mL, not 5ml
- I think D10 should be with 10 in subscript (D_{10})
- phyla is plural for phylum; verify/correct everywhere (sometimes phyla is used where it should be phylum, and vice versa)
- the authors use, in regard to IR, the terms 'tolerance', 'low-tolerance', 'intolerance', and 'sensitive' (oddly, the term 'resistance' in regard to IR is fully ignored though this term is often enough used in literature, including references 15, 23, 24, and 25). I suggest that the authors for this manuscript strictly adhere to 'tolerant' versus 'sensitive'.
- throughout the manuscript a tolerant/sensitive cutoff D_{10} at 200 Gy is used, yet in the first sentence of the abstract it reads "...radiosensitive bacteria succumbing to acute doses around 100Gy..". This is stated without reference (which is perhaps fine in an abstract) but is not repeated for instance in the discussion section, and there is simply no reference to it. Also, it is the only sentence in which dose has been qualified by the term 'acute' (everywhere else in the text, dose is unqualified ergo no distinction is been made between acute and chronic (or cumulative) doses. Reports on irradiation experiments should always include doserate and exposure time (together determining dose) as well as type and source of radiation. I suggest to rewrite the abstract, particularly the beginning, omitting the unreferenced 100 Gy mark (or write it differently, with reference), and to include in the first paragraph of the Discussion some aspects of doserate, dose, radiation type, and D_{10} value.
- the authors use 120 proteomes of 61 species with known D_{10} for TolRad training and testing. The idea is sound. However, I wonder whether all D_{10} 's refer to the same type of radiation? This might need checking. Probably, most D_{10} 's are for gamma's coming from Co-60 or Cs-137 (with specific energies), and presumably comparable doserates were used, yet some of these D_{10} 's may be derived from non-gamma i.e. X-ray or ions. In fact, the presented study determines D_{10} 's for two bacterial species using an X-ray irradiator (although keV range was not mentioned - see specific remark below). Though this may not influence the validity of the study or the discriminative power of TolRad, at least not significantly, I invite the authors to address these matters in the Discussion. The current text does not even refer once to gamma's in regard to survival, D_{10} 's, TolRad, doserates, tolerance, or sensitivity.

- as the authors point out (L453 and onwards) the train/test set is rather limited in size. This is in part due to the strict criterium of D_{10} determination (L189-190: "Only D_{10} s determined in liquid media or PBS and exposed at room temperature were included."). Two remarks: (1) to what extent was sporulation/germination taken into account (e.g. Bacillus, Clostridium, ..), and (2) why is the cyanobacterial clade so vastly underrepresented (only 1 representative?) – literature points to a number of IRR cyanobacteria, most notably extreme IR tolerant *Chroococcidiopsis* (approximated D_{10} value of 5 kGy of X-rays), *Anabaena* and *Limnospira* (*Arthrospira*) (both easily surviving 2-3 kGy), and moderate IR tolerant *Anacystis* (D_{10} value of 257 Gy of ^{60}Co γ radiation). Many if not all cyanobacteria are expected to have D_{10} around 100-200 Gy, because these bacteria have many cellular defenses in place to cope with ROS. Would be interesting to identify true IR sensitive cyanobacteria. At any rate, I think the authors need to delve into literature for an update on cyanobacterial IRR. By the way, I am aware that many cyanobacteria are multicellular (*Anabaena*, *Limnospira*,..) hampering classic D_{10} determinations.

- of the four predictors of the final TolRad model (L152-53, L377-405 and Table 2), only two seem to work well, while two other predictors work poorly? I don't know whether this is an acceptable result. Nonetheless, the (accuracy of the) prediction remained acceptably high. And the authors do express their concern and necessity for additional analysis (L402-5, L470-2).

- does the train/test set represent 5 or 7 phyla? (L191 versus L456 are discrepant)

- I understand TolRad being referred to by the authors as a model, but it is actually presented as a tool to discriminate/classify bacteria, with a certain degree of predictive accuracy, as being tolerant or sensitive to ionizing radiation (I guess if all DY's of all species in the training set are for gamma's, one should speak instead of "tolerant or sensitive to gamma radiation"). But the authors do not point to ready-to-use software or code or some sort of pipeline or GUI. How for instance can I test TolRad on the proteome(s) of my interest, i.e., input = proteome, output = class...? (not that I need to for this review)

- Train/Test set was not made available (e.g. together with the above on GitHub)

- TolRad as a tool could not be tested by me in a simple, user-friendly way

- references Aridhi *et al.* (2016), Vishambra *et al.* (2017), and Ryabova *et al.* (2020) should be included (see remarks further below)

- supplementary tables S1-S5 should be provided in flat file format

Specific:

L24: this is the only sentence in the entire manuscript where the word 'acute' is used. Qualifying a dose is fine, of course, but then it better needs to be done consistently. Especially in regard to biological systems (with a genetic response, damage repair, etc) when survival from chronic and acute exposure may be different, even at the same final dose. I suggest to clarify radiation quality in the Discussion and be careful to use undocumented qualifying terms in regard to dose.

L24: as said above, the cutoff at 100 Gy does not correspond with the rest of the manuscript, beginning just a few lines further, at L31

L27: something amiss with "The taxonomy-level diversity of IR of intolerance.."; also, the term intolerance is used, while I think it is better to adhere to two terms only i.e. tolerance and sensitivity; I suggest for this study to speak only of IR tolerant and IR sensitive bacteria.

L32: my first thought here was “why those 152 proteomes?”. Only on L243 it becomes clear that they corresponded to bacterial species previously detected in the human microbiome. My feeling is that this information needs to be mentioned in the abstract

L32: the term “hidden species” may not be understood

L37: replace phyla by phylum? Throughout the text verify contextual correctness of plural versus singular sense

L38: use Metagenome-Assembled Genomes if you refer to MAGs

L41: further **phyla**

L45-47: please rephrase “The ability..”; various experimental methods exist and have been used to elucidate bacterial traits in a HT fashion; and some computational ones too; please check literature for an update

L48: must be “to ionizing radiation”; in fact, the sentence should be “..to predict a bacterium as being tolerant or sensitive to ionizing radiation (IR).”

L49: yet another term: low-tolerance (...)

L51: the sentence “..that are of interest for both mechanistic studies into bacterial sensitivity to IR and biomarkers of IR exposure.” Makes no sense to me – rephrase please?

L71: “—“should not be used; by the way, what follows after “—“ is badly phrased

L86: “AlphaFold2” as this concerns the improved version presented by John Jumper in 2020 at CASP14.

L93-95: can you please check literature whether this is still true today for other abiotic stresses such as desiccation, temperature, salinity, pH, pressure, ..? In fact, are you aware that there is a strong link between desiccation and IRR? (see WoS or PubMed)

L128: better is “.. has only described 14 bacterial species with..” (& is ‘strains’ instead of ‘species’ perhaps more correct?)

L144: “that haver” should be “that have”

L143-5: problematic sentence, requires attention

L154-7: problematic sentences, requires attention

L161 replace intolerance by sensitivity?

L170: Trivedi et al., 2020 should be ref. 46 (CAUTION: this may cause reference renumbering)

L177: replace “bacterial stress tolerance” by “IR tolerance” or perhaps even “IR tolerance/sensitivity”?

L189-190: Only D₁₀ **values** determined in liquid media or PBS and exposed at room temperature were included.” What is the reason for using these criteria? I think you must explain this choice. In fact, nothing is said about spores (none in liquid standard conditions?), dose rate, exposure time, radiation quality.

L233: use “ five (out of 40, or 12.5%)”, not “5 (12.5%)”

L348: use something else instead of “ripe”, or rephrase sentence

L353: maybe better: “..to predict whether a bacterium might be sensitive or tolerant to IR (corresponding to a D_{10} respectively below or above 200 Gy).”

L355: URL unavailable

L372-4: please see:

- Aridhi et al., (2016). Prediction of Ionizing Radiation Resistance in Bacteria Using a Multiple Instance Learning Model. *J. Comp. Biology* 23(1):10-20. [<https://doi.org/10.1089/cmb.2015.0134>]
- Vishambra et al., (2017). Subcellular localization based comparative study on radioresistant bacteria: A novel approach to mine proteins involved in radioresistance, *Comp. Biol. and Chem.* 69:1-9. [<https://doi.org/10.1016/j.compbiolchem.2017.05.002>]
- Ryabova et al., (2020). DetR DB: A Database of Ionizing Radiation Resistance Determinants. *Genes* 11:1477 [<https://doi.org/10.3390/genes11121477>]

L370: manganese and iron do not need to start with capital?

L393: sentence seems garbled – please correct

L395: the “however” at the end is confusing

L430: taxology should be taxonomy, unless the IRS (the US Department) is involved :-)

L434: perhaps stick to tolerance versus sensitivity (tolerant versus sensitive)? (disuse intolerance)

L435-6: sentence is garbled, needs attention

L451: don't use the term intolerance..? I find sentence L450-1 a bit weird “Identifying environments that are rich in radiosensitive species can aid in understanding the biological causes of IR intolerance.” Is the biological cause not simply the lack (for the evolutionary need) of specialized repair and defense mechanisms, or do you think IR sensitive bacteria are actually the result of evolutionary loss, as early Earth was more IR-intensive (less ozone for protection, more natural radioactivity, ..)? Is this study about finding the ‘cause’ for IR sensitivity, or rather classifying bacteria as IR tolerant or IR sensitive? Maybe this paragraph deserves another ending?

L455: I suppose RT stands for room temperature – perhaps better to use it in full

L462: disuse the term intolerance..; phylum should be phyla

L473-4: see my remark for L372-4; change accordingly

L475-480: “We further demonstrate..training models.” Please rewrite alinea. You do not identify bacteria, you predict which bacteria might be IR sensitive/tolerant. *B. thetaiotaomicron* is not tolerant, it is sensitive ($D_{10} = 110$ Gy) (L419-20). Nothing is said about IRR *A. baumannii*? ($D_{10} = 400$ Gy)?

L490: come to think of it, perhaps rather use IR sensitive instead of radiosensitive, and IR tolerant instead of radiotolerant, etc. (although the use of radio as a prefix is probably generally accepted)

L498: approx. 70/30?

L509: should be improved e.g. “..to bin bacteria predicted as tolerant or sensitive to IR using the relative frequency of each of the predictor Pfam domains.”

L510: should probably be 10x (not 10X) or also “ten-fold”

L518: which version of UniProt? Possibly also give the date of the downloaded files? Something is amiss with the sentence “In Supplemental Table 4, UniProt Species names along with the species name the classification, the total number of missing Pfam IDs and taxonomy as well as the source suggesting that the species is in the HMB.” Please correct.

L522-536: I wonder whether not all MAGs are simply available via MGnify (EMBL) (<https://www.ebi.ac.uk/metagenomics>)

L525: should EMBLE not be EMBL? (better even EMBL-EBI); there is something wrong with this sentence, please correct

L530: replace “SRA (<https://www.ncbi.nlm.nih.gov/bioproject/>)” by “Sequence Read Archive (SRA) (<https://www.ncbi.nlm.nih.gov/sra/>)”; it occurs to me that NCBI never was introduced in the text (should have been written at first time use as “The National Center for Biotechnology Information (NCBI)”), i.e. on L321 as “.. were binned to 46 taxonomic clades according to the Taxonomy Database of the National Center for Biotechnology Information (NCBI) (Schoch et al., 2020) (<https://doi.org/10.1093%2Fdatabase%2Fbaaa062>), and 18 of these clades included multiple MAGs.”

L532: assemblies? Please correct sentence.

L533: which version of EggNOG-Mapper was used (5.0?)

L539: should be “was grown in”; what is the commercial source of BHIM? Reference?

L545: LB stands for Luria Broth or Luria-Bertani? Commercial source? Reference?

L545: “at 37” should be “at 37 °C”

L548-54: the authors give a dose rate (5 Gy/min?) but not an exposure time: which timepoints were used for the survival curves (Fig. 2B and C)? What were the corresponding doses? Please specify the keV-MeV range of the generated/applied X-rays. Concerning Fig. 2C: is the last datapoint about 180 Gy? Why? How was the extrapolation towards D₁₀ executed? A cumulative dose of 400 Gy could be easily achieved after 80 min?

L551: X-rays are waves, not particles, the aluminium filter blocks long- wavelength (low-energy) radiation ensuring a more uniform high-energy delivery (‘beam-hardening’). It occurs to me that 0.5 mm aluminium is not much for this apparatus – is this correct? I think for a 225 Type MultiRad with a 2.5 mm Al filter might deliver perhaps in the 5 Gy/min range. Please check technical specifications. Of course, applied voltage, beam angle, and exposure distance are also critical parameters. In order for researchers to be able to repeat or compare experiments, Methods must be unambiguously clear.

L559: why Sham with a capital? Just “sham” is fine. Some readers may not know what sham or sham-irradiation is in regard to the experimental and control sample.

Legend to Fig. 1: all species names in italic (including *Campylobacter coli*, *Morganella morganii*, etc.); and only names in italic, not strain alphanumeric codes or sp. notations etc.

Legend to Fig. 2: start with “Application of TolRad to...”; at L771: use “..by phylum.” (not by phyla); for 2B mention triplicates

Legend to Fig. 4: L798: use “..by phylum.” (not by phyla)

Legend to Suppl. Fig. S2: L810: use “..by phylum.” (not by phyla)

Fig. 1A: I believe the cutoff between moderately (gray) and extreme tolerant (blue) has not been discussed as such in the text? What is the cutoff? 2000 Gy?

Editor Comments

- Table 5, the "ATCC ID" is not correct for either of the strains listed, but instead you have the strain designation listed. I would suggest listing both, first by changing the header of the table to "strain designation" for the existing information, and then adding a third column for ATCC ID (which would be ATCC 29418 and ATCC 19606 for the respective strains of *B. thetaiotaomicron* and *A. baumannii* listed).
 - **RESPONSE:** Thank you for catching that switching of the strain designation and the ATCC ID. Table 5 has been corrected as suggested.

Species	Strain designation	ATCC ID
Bacteroides thetaiotaomicron	VPI 5482	ATCC 29418
Pseudomonas aeruginosa	PA103	ATCC 29260
Acinetobacter baumannii	2208	ATCC 19606

- The figure axis labels in Fig 2A appear to have been swapped (phylum/count).
 - **RESPONSE:** Thank you for noticing that mistake. The labels for Figure 2A have been corrected.

Reviewer # 1

- The authors briefly mentioned that Mn/Fe had been used as a predictor of a species' tolerance for IR. Based on their big data analysis, the authors should give more discussions regarding the **genetic and biomedical mechanisms for bacterial IR tolerance**.
 - **RESPONSE:** Thank you for this suggestion about further discussion on the genetic and biomedical mechanisms of IR tolerance. A section of the Discussion, ***Insights into the genetic traits of radiosensitivity (L393-425)*** does address the genetic mechanism of IR tolerance uncovered by the analysis. Specifically, each Pfam domains (i.e. genetic unit) used by the model is explored in the context of IR tolerance. An additional line has been added on the lack of metal binding domain in the Pfam domains utilized by TolRad to predict tolerance for IR *“Interestingly, none of the Pfam domains utilized by TolRad to predict IR tolerance involve iron-binding domains or metal ion importer domains, as may have been expected given the correlation between IR tolerance and the intracellular ratio of Mn/Fe”*. (L419-421). We hope this addition addresses the reviewer's concerns.

Reviewer # 2

General (for text including legends):

1. the abstract lacks emphasis on the importance/relevance of predicting radiosensitivity; I am thinking in terms of gut microbial dysbiosis during pelvic radiotherapy or changes in

human microbiomes under harsh space conditions, or the deployability of bacteria in radiation-intensive or radioactively-contaminated environments.

- **RESPONSE:** Thank you for this suggestion about the Abstract. The reviewer notes a lack of emphasis on the importance of predicting radiosensitivity in the abstract. The Importance section of the document **L44-L52** highlights the current lack of tools for the task of identifying IR-sensitive species and the utility that identifying more species would provide in identifying species that may be susceptible to IR exposure or that could be used for biomarker discovery. We have added an additional line on specific examples of radiation exposure are provided in the Importance section of the Abstract (**L98-L104**). "This model allows for the rapid screening of bacterial communities for IR-sensitive species that are of interest for studying the *mechanistic of bacterial sensitivity to IR and for identifying species that may host IR-responsive biomarkers.*" We hope this addresses the reviewer's concerns.
2. make sure all hyperlinks are active i.e. point to functional webpages
 - **RESPONSE:** The reviewer suggests checking the hyperlinks. The hyperlink for TolRad wasn't publicly live, this has been corrected.
 3. use SI units only i.e. mL, not ml
 - **RESPONSE:** *Thank you for noticing that typo; it has been corrected throughout the methods section.*
 4. always use a space between a given value and a unit i.e. 5 Gy, not 5Gy, or 5 mL, not 5ml
 - **RESPONSE:** *Thank you for noticing that typo; it has been corrected throughout the document section.*
 5. I think D10 should be with 10 in subscript (D₁₀)
 - **RESPONSE:** *Thank you for suggesting this formatting change; it has been made throughout the document.*
 6. phyla is plural for phylum; verify/correct everywhere (sometimes phyla is used where it should be phylum, and vice versa)
 - **RESPONSE:** *Thank you for noting these incorrect usages of phyla, they have been corrected throughout the document.*
 7. the authors use, in regard to IR, the terms 'tolerance', 'low-tolerance', 'intolerance', and 'sensitive' (oddly, the term 'resistance' in regard to IR is fully ignored though this term is often enough used in literature, including references 15, 23, 24, and 25). I suggest that the authors for this manuscript strictly adhere to 'tolerant' versus 'sensitive'.
 - **RESPONSE:** Thank you for this suggestion. The Review notes the literature's use of "resistance" instead of tolerance. In this paper, tolerance is used because we aren't referring to species with resistance to radiation, but rather those that can simply tolerate it. On the reviewers suggestion, the document has been changed as follows. The term "IR tolerant" and "IR sensitive" has been used where speaking generally species tolerance for radiation, and the term "radiosensitive" has been reserved for referring to species with a D₁₀ < 200Gy or species classified by TolRad as having a D₁₀ < 200Gy. Examples include; **L49, L116, L127, L132, L143, L161.**

8. throughout the manuscript a tolerant/sensitive cutoff D10 at 200 Gy is used, yet in the first sentence of the abstract it reads "...radiosensitive bacteria succumbing to acute doses around 100Gy..". This is stated without reference (which is perhaps fine in an abstract) but is not repeated for instance in the discussion section, and there is simply no reference to it. Also, it is the only sentence in which dose has been qualified by the term 'acute' (everywhere else in the text, dose is unqualified ergo no distinction is been made between acute and chronic (or cumulative) doses. Reports on irradiation experiments should always include doserate and exposure time (together determining dose) as well as type and source of radiation. I suggest to rewrite the abstract, particularly the beginning, omitting the unreferenced 100 Gy mark (or write it differently, with reference), and to include in the first paragraph of the Discussion some aspects of doserate, dose, radiation type, and D10 value.

- **RESPONSE:** Thank you for this feedback regarding discussion of radiation exposure in the document.
 - i. First, the reviewer notes a lack of clarity in the abstract about the use of the term "radiosensitive." the abstract has been updated to reads "with some species succumbing to acute doses as low as 60 Gy" to avoid confusion (**L24**).
 - ii. *Second, the reviewer notes a lack of emphasis on acute vs chronic doses. It is the authors' understanding that D_{10} values are traditionally calculated using acute doses and so only acute doses are discussed in this paper. Clarity has been added to **Introduction (L109, L110)** the **Results section Collection of the Train/Test Set** "For all D_{10} 's included in the Train/Test, all exposures were acute and conducted using gamma or x-ray sources." (**L192**) and **Discussion (L455)**, and the **methods (L512)**. We hope this address the reviewers concerns.*
 - iii. Third, *the reviewer notes a lack of "exposure times"*. The author agrees that this information is valuable but was not available in the majority of paper used for training TolRad, as most papers are conducting a wide range of doses as part of a survival curve.

9. the authors use 120 proteomes of 61 species with known D10 for TolRad training and testing. The idea is sound. However, I wonder whether all D10's refer to the same type of radiation? This might need checking. Probably, most D10's are for gamma's coming from Co-60 or Cs-137 (with specificenergies), and presumably comparable doserates were used, yet some of these D10's may be derived from non-gamma i.e. X-ray or ions. In fact, the presented study determines D10's for two bacterial species using an X-ray irradiator (although keV range was not mentioned - see specific remark below). Though this may not influence the validity of the study or the discriminative power of TolRad, at least not significantly, I invite the authors to address these matters in the Discussion. The current text doesnot even refer once to gamma's in regard to survival, D10's, TolRad, doserates, tolerance, or sensitivity.

- **RESPONSE:** Thank you for this question about the nature of the exposures included in the Train/Test set. The reviewer notes the different sources of ionizing radiation used in the field. As noted in the method, all exposures

conducted as part of this study were conducted using an X-ray source (L586). For the train/test set, all sources of ionizing radiation were included (gamma from Co-60 or Cs-137 as well as x-ray). While the author acknowledges that some differences in D_{10} have been noted between X-ray and Gamma sources, the source should not so drastically change the D_{10} so as to change the classification of the species as radiosensitive or tolerant. An additional note of the variation in D_{10} due to source has been added to the Limitations section of the discussion. “Additionally, there are multiple laboratory sources of IR (i.e. x-ray, Co60 and C137), which are regularly grouped; despite having variable effects on biological systems” (L478-481). We hope this addition to the Discussion address the reviewers concerns.

10. as the authors point out (L453 and onwards) the train/test set is rather limited in size.

This is in part due to the strict criterium of D_{10} determination (L189-190: “Only D_{10} s determined in liquid media or PBS and exposed at room temperature were included.”). Two remarks: (1) to what extent was sporulation/germination taken into account (e.g. Bacillus, Clostridium, ..), and (2) why is the cyanobacterial clade so vastly underrepresented (only 1 representative?) – literature points to a number of IRR cyanobacteria, most notably extreme IR tolerant Chroococciopsis (approximated D_{10} value of 5 kGy of X-rays), Anabaena and Limnospira (Arthrospira) (both easily surviving 2-3 kGy), and moderate IR tolerant Anacystis (D_{10} value of 257 Gy of ^{60}Co radiation). Many if not all cyanobacteria are expected to have D_{10} around 100-200 Gy, because these bacteria have many cellular defenses in place to cope with ROS. Would be interesting to identify true IR sensitive cyanobacteria. At any rate, I think the authors need to delve into literature for an update on cyanobacterial IRR. By the way, I am aware that many cyanobacteria are multicellular (Anabaena, Limnospira,..) hampering classic D_{10} determinations.

- **RESPONSE:** Thank you for these additional questions about the bacteria included in the train/test set and the applications of ToIRad to Phylum with variable tolerances for IR.
 - The reviewer first asks about other bacterial considerations to the Train/Test Set. Bacterial spores were not included, nor were desiccated or otherwise “inactive” states. Only metabolically active cultures were included.
 - Second, the reviewer asks about the lack of cyanobacterial in the Train/Test Set. During the generation of the Train/Test Set, the author attempted to find as many studies as possible that included clearly calculated D_{10} values using an agreeable methodology, and none of the identified species ended up being cyanobacteria. The author is unsure how to include cyanobacteria in this paper, which is about a model that makes predictions based on D_{10} values, when the reviewer acknowledges that calculating D_{10} values for Cyanobacteria is complicated by these species' multi-cellular nature. However, the author notes that in **Supplemental Table 4**, the Deep Sea MAGs include a sample identified as a Cyanobacteria and ToIRad classified

this organism as Radiosensitive. The focus of this paper is the development of TolRad and providing examples of how it can be applied to various genomes. As such, we invite future researchers to apply TolRad to organisms of interest to them, such as Cyanobacteria.

- Finally, toward the question of how can TolRad differentiate between species within a phylum that have different tolerance for IR, while we were not able to culture and test multiple species of Cyanobacteria in the response time, we did include another proteobacteria (*Pseudomonas aeruginosa*, Figure 2D) that TolRad predicted was radiosensitive, to compliment the existing proteobacteria we that we validated as being IR-tolerant (Figure 2C). We hope this addresses the reviewers concern about the ability of the model to differentiate between more closely related species with different tolerances for IR.
11. of the four predictors of the final TolRad model (L152-53, L377-405 and Table 2), only two seem to work well, while two other predictors work poorly? I don't know whether this is an acceptable result. Nonetheless, the (accuracy of the) prediction remained acceptably high. And the authors do express their concern and necessity for additional analysis (L402-5, L470-2).
- **RESPONSE:** *The reviewer notes that two of the predictors contribute less to the model; this is true, but the removal of either predictor reduces the ability of the model to classify the test set correctly, so all four were maintained in the model.*
12. does the train/test set represent 5 or 7 phyla? (L191 versus L456 are discrepant)
- **RESPONSE:** *Thank you for noticing this mistake, 7 Phyla are included in the Test/Train set. This has been corrected in the document. (L480)*
13. I understand TolRad being referred to by the authors as a model, but it is actually presented as a tool to discriminate/classify bacteria, with a certain degree of predictive accuracy, as being tolerant or sensitive to ionizing radiation (I guess if all DY's of all species in the training set are for gamma's, one should speak instead of "tolerant or sensitive to gamma radiation"). But the authors do not point to ready-to-use software or code or some sort of pipeline or GUI. How for instance can I test TolRad on the proteome(s) of my interest, i.e., input = proteome, output = class...? (not that I need to for this review)
- **RESPONSE:** *The reviewer asks about the ability of researchers to use TolRad to make their own classifications. By an oversight of the author, the Github hosting of TolRad was not made public. The r script housing TolRad can now be downloaded at <https://github.com/philipjsweet/TolRad> . Instructions for usage have been added to the methods section. *The random forest model trained and validated in this paper is the bases for the TolRad R-package, which has been uploaded to GitHub (<https://github.com/philipjsweet/TolRad>) along with a user guide. Briefly, the user provides a path to a folder with a collection of Pfam genome annotations of bacterial species for which radiation tolerance classifications are desired and the function outputs a dataframe with the classification of each species. (L550-560)**

14. Train/Test set was not made available (e.g. together with the above on GitHub)
 - **RESPONSE:** The Train/Test set is available in **Sup. Table 3** but on the reviewers suggestion will also be added to the GitHub
15. TolRad as a tool could not be tested by me in a simple, user-friendly way
 - **RESPONSE:** Apologies, the tool is publicly available as a nR-package R on github. (see above)
16. references Aridhi et al. (2016), Vishambra et al. (2017), and Ryabova et al. (2020) should be included (see remarks further below)
 - **RESPONSE:** *Addressed below.*
17. supplementary tables S1-S5 should be provided in flat file format
 - **RESPONSE:** Files will be provided as such on the GitHub but will remain in the current format within the paper for ease of human reading. We hope this addresss the reviewers concerns.

Specific:

1. L24: this is the only sentence in the entire manuscript where the word 'acute' is used. Qualifying a dose is fine, of course, but then it better needs to be done consistently. Especially in regard to biological systems (with a genetic response, damage repair, etc) when survival from chronic and acute exposure may be different, even at the same final dose. I suggest to clarify radiation quality in the Discussion and be careful to use undocumented qualifying terms in regard to dose.
 - **RESPONSE:** The review notes the difference between acute and chronic does. For clarity, all doses in the Train/Test set were acute and so predictions by TolRad are for acute doses. This question has been addressed in **General Comment 8** but clarity has been added to the follwoign lines, **L109, L110, L190, L454 and, L511**. We hope this addresses the reviewers concerns.
2. L24: as said above, the cutoff at 100 Gy does not correspond with the rest of the manuscript, beginning just a few lines further, at L31
 - **RESPONSE:** This question has been addressed in **General Comment 8**
3. L27: something amiss with "The taxonomy-level diversity of IR of intolerance.."; also, the term intolerance is used, while I think it is better to adhere to two terms only i.e. tolerance and sensitivity; I suggest for this study to speak only of IR tolerant and IR sensitive bacteria.
 - **RESPONSE:** This question has been addressed in **General Comment 8**
4. L32: my first thought here was "why those 152 proteomes?". Only on L243 it becomes clear that they corresponded to bacterial species previously detected in the human microbiome. My feeling is that this information needs to be mentioned in the abstract
 - **RESPONSE:** This comment has been clarified in the abstract (**L33**).
5. L32: the term "hidden species" may not be understood
 - **RESPONSE:** On the reviewer's suggestion, the abstracted has been changed to say "untrained" instead of hidden (**L31**).
6. L37: replace phyla by phylum? Throughout the text verify contextual correctness of plural versus singular sense
 - **RESPONSE:** Document was checked to ensure proper use of phyla/phylum

7. L38: use Metagenome-Assembled Genomes if you refer to MAGs
 - **RESPONSE:** corrected in the document
8. L41: further phyla
 - **RESPONSE:** corrected in the document
9. L45-47: please rephrase “The ability..”; various experimental methods exist and have been used to elucidate bacterial traits in a HT fashion; and some computational ones too; please check literature for an update
 - **RESPONSE:** Changed to state, “*The ability to harness the full potential of bacterial diversity is hampered by the limited number of high-throughput experimental or bioinformatic methods for characterizing bacterial traits*” (L46-47)
10. L48: must be “to ionizing radiation”; in fact, the sentence should be “..to predict a bacterium as being tolerant or sensitive to ionizing radiation (IR).”
 - **RESPONSE:** On the reviewer advice, the document was changed to “*bacteria as IR tolerant ($D_{10} > 200\text{Gy}$) or radiosensitive ($D_{10} < 200\text{Gy}$)*” (L48-49)
11. L49: yet another term: low-tolerance (...)
 - **RESPONSE:** *On the reviewer advice, the document was changed to “IR-sensitive species”* (L50)
12. L51: the sentence “..that are of interest for both mechanistic studies into bacterial sensitivity to IR and biomarkers of IR exposure.” Makes no sense to me – rephrase please?
 - **RESPONSE:** *Clarified as “mechanistic studies of bacterial sensitivity to IR and for identifying IR-responsive biomarkers.”* (L51)
13. L71: “—“should not be used; by the way, what follows after “—“ is badly phrased
 - **RESPONSE:** *The review took issue with the phrasing of Line 71, it has been changed to “eliminating a barrier that previously limited the”* (L72)
14. L86: “AlphaFold2” as this concerns the improved version presented by John Jumper in 2020 at CASP14. (L87)
 - **RESPONSE:** *Thank you for this remark, the source and text has been corrected.*
15. L93-95: can you please check literature whether this is still true today for other abiotic stresses such as dessication, temperature, salinity, pH, pressure, ..? In fact, are you aware that there is a strong link between dessication and IRR? (see WoS or PubMed)
 - **RESPONSE:** The author meant to refer only to the ability to predict stress tolerance computationally. This has been updated in the text. . *The ability to computationally predict bacterial tolerance for stress from such genome annotations is currently limited to antibiotic resistance* (L94)
16. L128: better is “.. has only described 14 bacterial species with..” (& is ‘strains’ instead of ‘species’ perhaps more correct?)
 - **RESPONSE:** *corrected in the document*
17. L144: “that haver” should be “that have”
 - **RESPONSE:** *corrected in the document*
18. L143-5: problematic sentence, requifres attention
 - **RESPONSE:** *corrected in the document*
19. L154-7: problematic sentences, requifres attention
 - **RESPONSE:** *corrected in the document* (L154)

20. L161 replace intolerance by sensitivity?
- **RESPONSE:** *Address in General Comment 8*
21. L170: Trivedi et al., 2020 should be ref. 46 (CAUTION: this may cause reference renumbering)
- **RESPONSE:** *corrected in the document (L168)*
22. L177: replace “bacterial stress tolerance” by “IR tolerance” or perhaps even “IR tolerance/sensitivity”?
- **RESPONSE:** *Address in General Comment 8*
23. L189-190: Only D10 values determined in liquid media or PBS and exposed at room temperature were included.” What is the reason for using these criteria? I think you must explain this choice. In fact, nothing is said about spores (none in liquid standard conditions?) , dose rate, exposure time, radiation quality.
- **RESPONSE:** The review asks about the selection conditions for the Test//Trainset. These conditions were selected because they represent control conditions, in contrast to experimental conditions (like being embedded in meat or kept on ice) that can change the D₁₀. Spores were excluded from the study as the focus was vegetative cells. The following changes were made to the text “*Due to the impact that exposure conditions can have on IR tolerance, only D₁₀'s determined in liquid media or PBS and exposed at room temperature were included. For all D10's included in the Train/Test, all exposures were acute and conducted using gamma or x-ray sources.*” (L189-192).
24. L233: use” five (out of 40, or 12.5%)”, not “5 (12.5%)”
- **RESPONSE:** *corrected in the document*
25. L348: use something else instead of “ripe”, or rephrase sentence
- **RESPONSE:** *Changed to “have potential”*
26. L353: maybe better: “..to predict whether a bacterium might be sensitive or tolerant to IR (corresponding to a D10 respectively below or above 200 Gy).”
27. L355: URL unavailable
- **RESPONSE:** The directory has been made publicly available (L387)
28. L372-4: please see:
- Aridhi et al., (2016). Prediction of Ionizing Radiation Resistance in Bacteria Using a Multiple Instance Learning Model. *J. Comp. Biology* 23(1):10-20. [[httpResponse: //doi.org/10.1089/cmb.2015.0134](https://doi.org/10.1089/cmb.2015.0134)].
 - **RESPONSE:** This article is behind a paywall and can't be accessed by the author to understand the scope of this mode, but the document has been updated to reflect the existence of other radiation tolerance predictive models highlighted by the reviewers. (L389, L506)
 - Vishambra et al., (2017). Subcellular localization based comparative study on radioresistant bacteria: A novel approach to mine proteins involved in radioresistance, *Comp. Biol. and Chem.* 69:1-9. [[httpResponse: //doi.org/10.1016/j.compbiolchem.2017.05.002](https://doi.org/10.1016/j.compbiolchem.2017.05.002)]
 - **RESPONSE:** This paper is has been referenced in the discussion (L399)
 - Ryabova et al., (2020). DetR DB: A Database of Ionizing Radiation Resistance Determinants. *Genes* 11:1477 [[httpResponse: //doi.org/10.3390/genes11121477](https://doi.org/10.3390/genes11121477)]

- **RESPONSE:** This paper is has been referenced in the discussion (**L399**)
29. L370: manganese and iron do not need to start with capital?
- **RESPONSE:** Capitalizing the first letter in the abbreviation of atoms is common practice.
30. L393: sentence seems garbled – please correct
- **RESPONSE:** the PFAM domain in question was clarified.
31. L395: the “however” at the end is confusing
- **RESPONSE:** The however was removed, and the sentence simplified to “The contribution of the PF03466, *LysR substrate binding domain*, which is often found in ligand response transcription factors, and it has been previously noted that while the radiosensitive bacterium *S. oneidensis* has 52 proteins in the LysR Family, the radioresistant bacterium *D. radiodurans* only has two.” (**L411-415**)
32. L430: taxology should be taxonomy, unless the IRS (the US Department) is involved :-)
- **RESPONSE:** *corrected in the document*
33. L434: perhaps stick to tolerance versus sensitivity (tolerant versus sensitive)? (disuse intolerance)
- **RESPONSE:** *corrected in the document*
34. L435-6: sentence is garbled, needs attention
- **RESPONSE:** *Sentence was changed from “we found a diversity of putative radiosensitive.” To “we found a diversity of putative radiosensitive species”*
35. L451: don’t use the term intolerance..? I find sentence L450-1 a bit weird “Identifying environments that are rich in radiosensitive species can aid in understanding the biological causes of IR intolerance.” Is the biological cause not simply the lack (for the evolutionary need) of specialized repair and defense mechanisms, or do you think IR sensitive bacteria are actually the result of evolutionary loss, as early Earth was more IR-intensive (less ozon for protection, more natural radioactivity, ..)? Is this study about finding the ‘cause’ for IR sensitivity, or rather classifying bacteria as IR tolerant or IR sensitive? Maybe this paragraph deserves another ending?
- **RESPONSE:** First, the reviewer notes the use of the term “*IR Intolerance*” as inconsistent with other sections. The sentence has been changed to “*IR sensitive*”. Next the reviewer asks about the implications of the need for identifying IR sensitive species. The author notes that not all species in the deep ocean samples were characterized as IR sensitive, so there appears to be more at play than just the immediate environmental pressures. This study is interested in finding species that are sensitive to IR that can be used for additional comparative studies, as has been performed with IR tolerant species isolated from different environments. We hope this address the reviewers concerns.
36. L455: I suppose RT stands for toom temperature – perhaps better to use it in full
- **RESPONSE:** *corrected in the document*
37. L462: disuse the term intolerance..; phylum should be phyla
- **RESPONSE:** *corrected in the document*
38. L473-4: see my remark for L372-4; change accordingly
- **RESPONSE:** *corrected in the document*

39. L475-480: "We further demonstrate..training models." Please rewrite alinea. You do not identify bacteria, you predict which bacteria might be IR sensitive/tolerant. B. thetaiotaomicron is not tolerant, it is sensitive (D10 = 110 Gy) (L419-20). Nothing is said about IRR A. baumannii? (D10 = 400Gy)?
- **RESPONSE:** Thank you for these clarifying questions about the experimental validation of the predications made by TolRad.
 - In response to the reviewer's comment, "*Identify novel bacteria*" was changed to "*identify putative bacteria*," and "*tolerance of B. thetaiotaomicron*: was changed to "*IR sensitivity of B. thetaiotaomicron*" (**L502**)
 - To address the reviewer's concerns about the experimental support for the predictive power of TolRad . *A.baumannii* was included as a control (predicted to be IR tolerant, demonstrated to be IR tolerant). The focus of the paper is the finding of radiosensitive species and so the *A.baumannii* result is not highlighted in the final summary paragraph. For additional support of the ability of TolRad to correctly differentiate radiosensitive species from species tolerant of IR exposure, we have included an additional experimental validation **Figure 2D** and **L272-277**.
40. L490: come to think of it, perhaps rather use IR sensitive instead of radiosensitive, and IR tolerant instead of radiotolerant, etc. (although the use of radio as a prefix is probably generally accepted)
- **RESPONSE:** Address in **General Comment 8**
41. L498: approx. 70/30?
- **RESPONSE:** *corrected in the document*
42. L509: should be improved e.g. "...to bin bacteria predicted as tolerant or sensitive to IR using the relative frequency of each of the predictor Pfam domains."
- **RESPONSE:** On the reviewers suggestion, the wording was changed from "*bin*" to "*classify*".
43. L510: should probably be 10x (not 10X) or also "ten-fold"
- **RESPONSE:** *corrected in the document* (**L537**)
44. L518: which version of UniProt? Possibly also give the date of the downloaded files? Something is amiss with the sentence "In Supplemental Table 4, UniProt Species names along with the species name the classification, the total number of missing Pfam IDs and taxonomy as well as the source suggesting that the species is in the HMB." Please correct.
- **RESPONSE:** On the reviewers suggestion, the date the UniProt annotations were downloaded was added to the document. Additionally, the following clarifications were added for the description of **Supplemental Table 4** "*all of the species on which TolRad radiation tolerance predictions were made can be found. For each entry, where applicable, the following is provided; UniProt Species names, ATTC strain ID (ATCC_Strain), the ATCC database name (ATCC_Name), the internal reference ID (ID), total number of missing predictor Pfam IDs (Missing Pfam's) and taxonomy (Phylum) as well as the literature*

source suggesting that the species is in the HMB (source), and the MAG ID.” The methods have also been clarified (L546-551)

45. L522-536: I wonder whether not all MAGs are simply available via MGnify (EMBL) ([httpResponse: //www.ebi.ac.uk/metagenomics](http://www.ebi.ac.uk/metagenomics))
- **RESPONSE:** *The reviewer notes that MAGs may be available on Mgnify, this is possible, but the author felt reporting the original source of the sequences used in this study was important and so the text was not changed.*
46. L525: should EMLE not be EMBL? (better even EMBL-EBI); there is something wrong with this sentence, please correct
- **RESPONSE:** *corrected in the document*
47. L530: replace “SRA ([httpResponse: //www.ncbi.nlm.nih.gov/bioproject/](http://www.ncbi.nlm.nih.gov/bioproject/))” by “Sequence Read Archive (SRA) ([httpResponse: //www.ncbi.nlm.nih.gov/sra/](http://www.ncbi.nlm.nih.gov/sra/))”; it occurs to me that NCBI never was introduced in the text (should have been written at first time use as “The National Center for Biotechnology Information (NCBI)”, i.e. on L321 as “.. were binned to 46 taxonomic clades according to the Taxonomy Database of the National Center for Biotechnology Information (NCBI) (Schoch et al., 2020) ([httpResponse: //doi.org/10.1093/database/Fbaaa062](http://doi.org/10.1093/database/Fbaaa062)), and 18 of these clades included multiple MAGs.”
- **RESPONSE:** *both were corrected in the document (L555 and L560)*
48. L532: assemblies? Please correct sentence.
- **RESPONSE:** *corrected in the document (L562)*
49. L533: which version of EggNOG-Mapper was used (5.0?)
- **RESPONSE:** *corrected in the document (L563)*
50. L539: should be “was grown in”; what is the commercial source of BHIM? Reference?
- **RESPONSE:** *The provider of BHIM (Fisher scientific) was added to the document. (L575)*
51. L545: LB stands for Luria Broth or Luria-Bertani? Commercial source? Reference?
- **RESPONSE:** *The meaning of LB (Luria Broth) and the provider of LB (Fisher scientific) were added to the document. (L577-578)*
52. L545: “at 37” should be “at 37 °C”
- **RESPONSE:** *corrected in the document (L571)*
53. L548-54: the authors give a dose rate (5 Gy/min?) but not an exposure time: which timepoints were used for the survival curves (Fig. 2B and C)? What were the corresponding doses? Please specify the keV-MeV range of the generated/applied X-rays. Concerning Fig. 2C: is the last datapoint about 180 Gy? Why? How was the extrapolation towards D10 executed? A cumulative dose of 400 Gy could be easily achieved after 80 min?
- **RESPONSE:** *The reviewer asks for specifics about the radiation exposure conditions. Exposure times can be calculated from the dose rate, i.e. a 10Gy dose would take 2 minutes. The mA and Kv of the exposure are provided in the original document (L550). Next, the reviewer asks about, timepoints, but timepoints were not collected as part of this experiment. After exposure, the samples were plated. No recovery was conducted. Next the reviewer asks about the failure to deliver 400Gy doses. The X-ray machine available at UT Austin*

cannot deliver such high doses. Additionally, at doses so close to the D10, the number of CFU's becomes more erratic. Lastly, the reviewer asks about how the D10 was collected. A logarithm line was fit to the collected CFU data and the point at which 10% would remain was calculated. This has been clarified in the methods (**L591**)

54. L551: X-rays are waves, not particles, the aluminium filter blocks long- wavelength (low-energy) radiation ensuring a more uniform high-energy delivery ('beam-hardening'). It occurs to me that 0.5mm aluminium is not much for this apparatus – is this correct? I think for a 225 Type MultiRad with a 2.5 mm Al filter might deliver perhaps in the 5 Gy/min range. Please check technical specifications. Of course, applied voltage, beam angle, and exposure distance are also critical parameters. In order for researchers to be able to repeat or compare experiments, Methods must be unambiguously clear.

- **RESPONSE:** The reviewers ask for more details about the exposure apparatus. The tunable parameters (mA and kV) are reported. Shelf height for these exposures was Stage 5 (29.5cm from source). The Reviewer asks about the filter used for exposure, and while many filters are available for the 225 MultiRad for all of the exposures reported in this paper, a 0.5mm Al filter was used. Shelf height has been clarified in the document "*When exposing cells to IR a Faxitron 225 MultiRad X-ray irradiator was used. The machine was set to 12 mA, 220 kV with a shelf height of 44.5cm for doses below 10Gy, at a height of 37cm for doses between 10-70Gy and a height of 29.5cm for doses above 70Gy. A 0.5 mm aluminum filter ensured that only high-energy x-rays were delivered. Exposures were conducted at room temperature (24 °C).*"

- (**L586-590**)

55. L559: why Sham with a capital? Just "sham" is fine. Some readers may not know what sham or sham- radiation is in regard to the experimental and control sample.

- **RESPONSE:** *The reviewer notes the lack of clarity in using "sham", this has been changed to "sham (0Gy)" (L595)*

Figures

1. Legend to Fig. 1: all species names in italic (including *Campylobacter coli*, *Morganella morganii*, etc.; and only names in italic, not strain alphanumeric codes or sp. notations etc.

- **RESPONSE:** *Thank you for pointing out that some species names were incorrectly formatted in Figure 1; this has been corrected.*

2. Legend to Fig. 2: start with "Application of TolRad to...";

- **RESPONSE:** *Thank you for noticing that typo; it has been corrected.*

3. L771: use "...by phylum." (not by phyla); for 2B, mention triplicates

- **RESPONSE:** *Thank you for noticing that omission; the number of replicates has been added to the Figure 2 caption.*

4. Legend to Fig. 4: L798: use "...by phylum." (not by phyla)

- **RESPONSE:** *Thank you for noticing that typo; it has been corrected.*

5. Legend to Suppl. Fig. S2: L810: use "...by phylum." (not by phyla)

- **RESPONSE:** *Thank you for noticing that typo; it has been corrected.*

6. Fig. 1A: I believe the cutoff between moderately (gray) and extreme tolerant (blue) has not been discussed as such in the text? What is the cutoff? 2000 Gy?

- **RESPONSE:** The reviewer asks about the distinction between extremely tolerant and tolerant species in Figure 1A. The cut-off was based on bacteria that have been previously described as highly tolerant of radiation and is described in **L198** as “D10 > 1500Gy”. In the original draft, there was not a citation for this line, but it is in reference to *Acinetobacter radioresistens*. Upon review, we noted that the D10 of *A. radioresistens* is in fact 1.2kGy, and the document has been updated to reflect this change.

Re: Spectrum03838-23R1 (ToIRad: A model for predicting radiation tolerance using Pfam annotations identifies novel radiosensitive bacterial species from reference genomes and MAGs)

Dear Dr. Lydia M Contreras:

Thank you for your patience while I reviewed your revised paper. You did an excellent job responding to all the reviewer concerns and my own as well. I have no further requirements for publication.

Your manuscript has been accepted, and I am forwarding it to the ASM production staff for publication. Your paper will first be checked to make sure all elements meet the technical requirements. ASM staff will contact you if anything needs to be revised before copyediting and production can begin. Otherwise, you will be notified when your proofs are ready to be viewed.

Sincerely,
Jonathan Jacobs
Editor
Microbiology Spectrum